# Customizing Visual Emotion Evaluation for MLLMs: An Open-vocabulary, Multifaceted, and Scalable Approach

**Daiqing Wu**[1,4]  **Dongbao Yang**[1,✉]  **Sicheng Zhao**[3]  **Can Ma**[1,✉]  **Yu Zhou**[2]

[1]Institute of Information Engineering, Chinese Academy of Sciences
[2]VCIP & TMCC & DISSec, College of Computer Science, Nankai University
[3]Department of Psychological and Cognitive Sciences, Tsinghua University
[4]University of Chinese Academy of Sciences

`wudaiqing@iie.ac.cn  yangdongbao@iie.ac.cn  macan@iie.ac.cn`

## Abstract

Recently, Multimodal Large Language Models (MLLMs) have achieved exceptional performance across diverse tasks, continually surpassing previous expectations regarding their capabilities. Nevertheless, their proficiency in perceiving emotions from images remains debated, with studies yielding divergent results in zero-shot scenarios. We argue that this inconsistency stems partly from constraints in existing evaluation methods, including the oversight of plausible responses, limited emotional taxonomies, neglect of contextual factors, and labor-intensive annotations. To facilitate customized visual emotion evaluation for MLLMs, we propose an Emotion Statement Judgment task that overcomes these constraints. Complementing this task, we devise an automated pipeline that efficiently constructs emotion-centric statements with minimal human effort. Through systematically evaluating prevailing MLLMs, our study showcases their stronger performance in emotion interpretation and context-based emotion judgment, while revealing relative limitations in comprehending perception subjectivity. When compared to humans, even top-performing MLLMs like GPT4o demonstrate remarkable performance gaps, underscoring key areas for future improvement. By developing a fundamental evaluation framework and conducting a comprehensive MLLM assessment, we hope this work contributes to advancing emotional intelligence in MLLMs. Project page: https://github.com/wdqqdw/MVEI.

## 1 Introduction

Perceiving emotional signals from visual stimuli is fundamental for humans to refine decision-making and build effective communication (Schutte et al., 2001), and modeling this capability has led to the emergence of Affective Image Content Analysis (AICA) as a key research direction (Zhao et al., 2022). Recently, the advent of Multimodal Large Language Models (MLLMs) has revolutionized image understanding tasks (Yang et al., 2023c). However, their competence in AICA remains contested. Divergent findings underscore a paradox: while some studies (Xie et al., 2024; Wu et al., 2025) demonstrate MLLMs' limited emotion recognition performance, others successfully employ them as emotion annotators for data augmentation (Lian et al., 2025; Cheng et al., 2024). We attribute this discrepancy to the incompatibility of conventional emotion evaluation approaches with MLLMs.

Specifically, current evaluation approaches can be broadly categorized into emotion classification and emotion interpretation, as illustrated in Figure 1 (a,b). In emotion classification, models are required to assign the affective state of an input image to a predefined set of emotion categories. Most benchmarks (You et al., 2016; Yang et al., 2023a) provide a single label per image, while a few (Kosti et al., 2017; Wei et al., 2020) incorporate multiple labels. In contrast, emotion interpretation focuses on understanding the underlying causes of emotions in images. It encompasses two primary sub-tasks: explaining the causes of emotional states (Achlioptas et al., 2021; 2023) and identifying salient visual elements that contribute to emotional responses (Lin et al., 2025).

When applied to MLLMs, these methods reveal four primary limitations. *Firstly*, their adoption of fixed ground-truth answers for open-ended questions imposes structural constraints that exclude

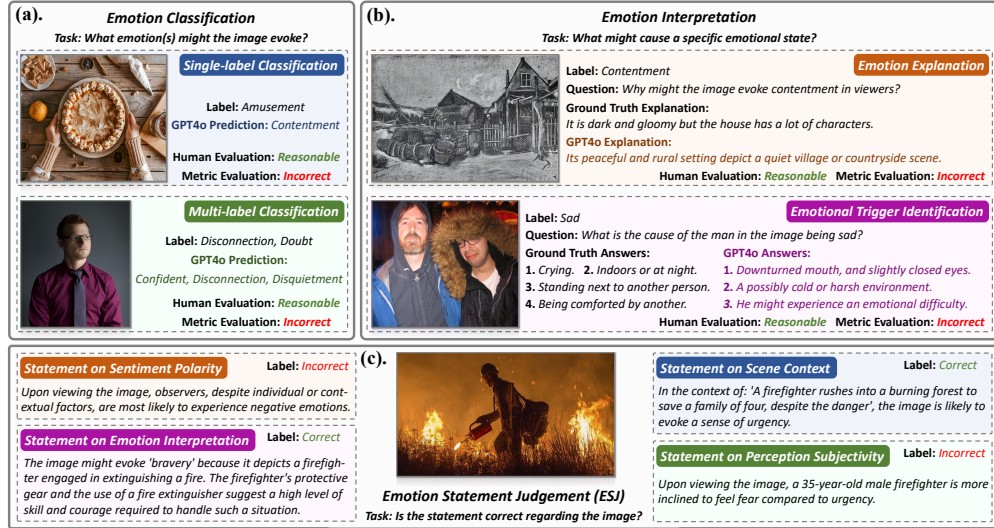

Figure 1: Comparison between current emotion evaluation approaches and the proposed ESJ.

other plausible responses. Emotion perception is inherently subjective (Zhao et al., 2016), as the same image may evoke divergent reactions across individuals, and emotional states permit varied interpretations. As demonstrated in Figure 1, responses generated by GPT4o (Hurst et al., 2024) that seem reasonable to humans are judged as inaccurate under rigid evaluation metrics. *Secondly*, they are mostly constructed upon emotion theories with limited emotional taxonomies. Popular emotion classification and interpretation benchmarks, such as FI (You et al., 2016) and Artemis (Achlioptas et al., 2021), comprise only eight emotion categories. Such taxonomic granularity fails to capture fine-grained affective variations between images. *Thirdly*, they focus solely on intrinsic image attributes while overlooking critical contextual dimensions. As recognized in established psychological literature, emotion perception can be influenced by extravisual factors (Barrett et al., 2011), including the scene context in which the image is set (Wieser & Brosch, 2012), as well as the viewer's identity and personality (Hamann & Canli, 2004). *Fourthly*, they predominantly rely on majority voting mechanisms to ensure label reliability in crowdsourced annotations (Li et al., 2017), which is labor-intensive, particularly for fine-grained annotation tasks. EMOTIC (Kosti et al., 2017), for instance, requires coordination with 23,788 annotators. This operational burden severely constrains dataset scalability in magnitude and generalization capacity across image domains.

To facilitate customized visual emotion evaluations for MLLMs, we propose a dual-component solution for these limitations: the Emotion Statement Judgment (**ESJ**) task, complemented by the **INSETS** (**IN**telligent Vi**S**ual **E**motion **T**agger and **S**tatement Constructor) pipeline for efficient annotation. In designing the framework, we emphasize evaluation precision over complexity to establish a reliable offline standard. With this aim, *ESJ reformulates visual emotion evaluation by requiring MLLMs to validate emotion-centric statements for a given image*. It effectively mitigates ambiguity in open-ended questions while being highly extensible for evaluation depth and diversity. In parallel, INSETS annotates images with multiple open-vocabulary emotion labels, significantly refining the emotional taxonomies. These labels are then utilized to construct multifaceted emotion-centric statements, covering both intrinsic image attributes and extrinsic contextual factors. Crucially, only minimal human intervention is required, ensuring a high scalability of the approach.

Leveraging INSETS, we automatically construct INSETS-462k, a large-scale annotated ESJ corpus. Building on it, we curate **MVEI** benchmark (**M**ultifaceted evaluation of **V**isual **E**motion **I**ntelligence) through careful human refinement. MVEI comprises 3,086 unique image–statement pairs designed to enable comprehensive evaluation of MLLMs. Grounded in established theories of affective cognition, it covers four complementary dimensions: sentiment polarity (Russell, 1980), emotion interpretation (Ekman & Friesen, 1971), scene context (Barrett et al., 2011), and perception subjectivity (Hamann & Canli, 2004). Systematic evaluation reveals that recent MLLMs exhibit considerable proficiency but still lag behind humans, particularly in discerning emotional polarity and interpreting perception subjectivity. Further explorations indicate that the former can likely be improved through targeted adaptation, whereas the latter is more tied to the models' inherent properties, highlighting potential directions for future research. In summary, the contributions of this paper are threefold:

- We identify four major limitations in existing visual emotion evaluations for MLLMs and introduce the customized Emotion Statement Judgement task to address them.

- Complementing the ESJ task, we further develop the INSETS pipeline, offering a scalable approach to annotating images with open-vocabulary emotion labels and constructing multifaceted emotion-centric statements with minimal human effort.

- Building on INSETS annotations with human refinement, we curate the MVEI benchmark, followed by a systematic evaluation of recent MLLMs. Comprehensive results provide insights and foster further advancements in visual emotional intelligence.

## 2 RELATED WORKS

### 2.1 AICA BENCHMARKS

Psychological researchers conceptualize emotion representation through two principal frameworks: the Categorical Emotion Space (CES), which discretizes affective states into predefined taxonomies, and the Dimensional Emotion Space (DES), which maps emotions onto continuous coordinations. For simplicity and better interpretability, most benchmarks adopt emotion classification evaluations based on discrete CES emotion taxonomies. This category encompasses both early small-scale benchmarks, such as IAPSa (Mikels et al., 2005) and Abstract (Machajdik & Hanbury, 2010), as well as later larger-scale benchmarks like FI (You et al., 2016) and WebEmo (Panda et al., 2018). Over time, benchmarks with enriched metadata have also been developed. Notable examples include EMOTIC (Kosti et al., 2017), which integrates multiple emotion categories, VAD values (Schlosberg, 1954), and human-related bounding boxes, and EmoSet (Yang et al., 2023a), which employs describable emotion attributes that cover different levels of visual information.

Some other benchmarks adopt emotion interpretation evaluations by extending CES-based taxonomies with additional emotional explanations, such as Artemis (Achlioptas et al., 2021) and Affection (Achlioptas et al., 2023). EIBench (Lin et al., 2025) diverges slightly, shifting focus on identifying visual emotional triggers. Based on these benchmarks, numerous expert models (Jia & Yang, 2022; Feng et al., 2023; Wu et al., 2024) have been developed, demonstrating strong performance under the fine-tuning and testing paradigm. In contrast, MLLMs are commonly pre-trained on web-scale data, without explicitly aligning with benchmark-specific knowledge. This discrepancy introduces multiple constraints when applying conventional benchmarks to MLLMs, necessitating customized visual emotion evaluation approaches that account for their generalized knowledge structures.

### 2.2 EVALUATION OF MLLMS

Recent years have witnessed growing academic and industrial interest in MLLMs. Unlike specialized models, MLLMs demonstrate versatile competence across diverse tasks (Li & Lu, 2024), fueling expectations for their trajectory toward Artificial General Intelligence (Maruyama, 2020). To evaluate MLLMs, various benchmarks have been established, covering perception (Li et al., 2023b; Liu et al., 2023), reasoning (Nie et al., 2024; Zhang et al., 2019), ethics (Qian et al., 2024b; Guan et al., 2024), and specialized domains (Chen et al., 2024a; Qian et al., 2024a). Yet emotional intelligence remains conspicuously underexplored, particularly in the visual modality. In existing efforts, MM-BigBench (Yang et al., 2023b) simply aggregates mainstream image-text benchmarks; FABA-Bench (Li et al., 2024) focuses primarily on facial expressions and actions; EmoBench-M (Hu et al., 2025) and EEmo-Bench (Gao et al., 2025), while largely extending task coverage, still insufficiently handle the ambiguity inherent in open-ended questions. To fill this gap, we propose the ESJ task and the MVEI benchmark, aiming to customize and advance visual emotion evaluation of MLLMs.

## 3 EMOTION STATEMENT JUDGEMENT

ESJ aims to evaluate the competence of MLLMs in perceiving emotions from visual content. In each trial, MLLMs receive an image and a paired emotion-centric statement. MLLMs are then tasked to judge whether the statement is accurate in relation to the image. To ensure both breadth and depth in evaluation, we draw inspiration from cognitive research (Shuman et al., 2017) and AICA surveys (Zhao et al., 2023), and design emotion-centric statements from four complementary

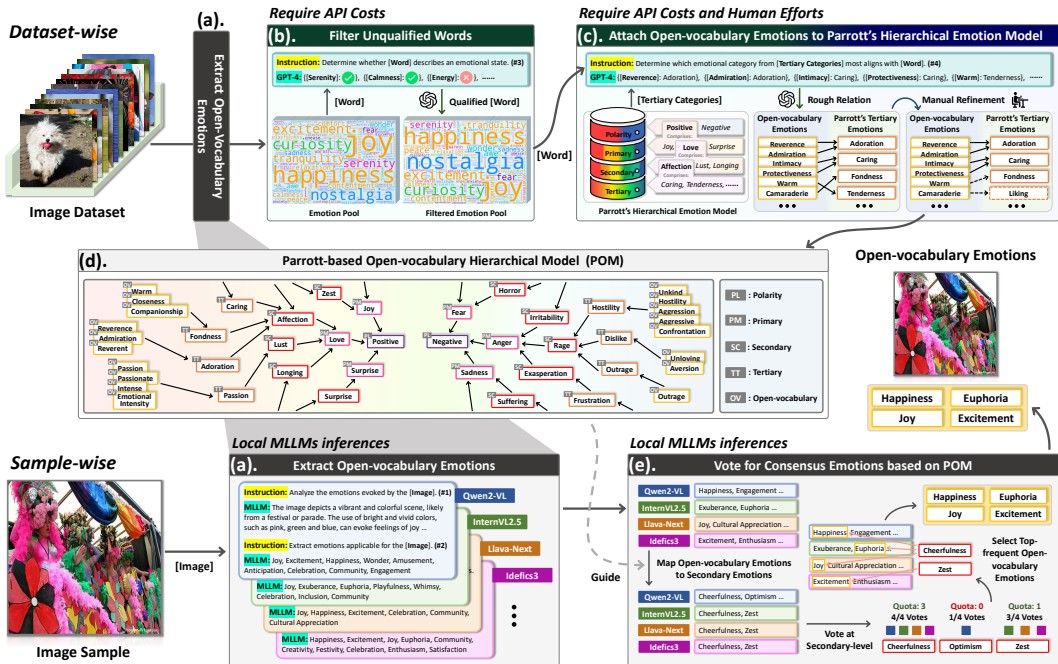

Figure 2: Illustration of the open-vocabulary emotion tagging stage. We first extract all potential open-vocabulary emotions from the image dataset (a) and then attach these emotions to a well-established emotion model (b,c). Through this model (d), we identify and select open-vocabulary emotions consistently recognized by multiple MLLMs as the labels of each image (e).

dimensions: **1).** *Sentiment Polarity Statements* require MLLMs to decide sentiment polarities without any additional clues, aiming to assess MLLMs' proficiency in directly identifying the basic emotional tone. **2).** *Emotion Interpretation Statements* ask MLLMs to verify the consistency between affective explanations and corresponding emotional states. They measure MLLMs' affective reasoning capability given specific emotional triggers. **3).** *Scene Context Statements* probe MLLMs' comprehension of the dynamic interplay between the potential scene context where the image takes place, and image-evoked emotional responses. **4).** *Perception Subjectivity Statements* task MLLMs to predict the personalized emotional responses under assumptions of specific viewer identities, examining whether MLLMs can recognize how subjectivity shapes emotional perceptions.

Collectively, these dimensions establish a holistic visual emotion evaluation framework for MLLMs. They cover both intrinsic image attributes emphasized in existing benchmarks and underexplored contextual factors critical for human emotional perception (Stemmler & Wacker, 2010).

# 4  ANNOTATION PIPELINE: INSETS

Complementing the ESJ task, we design an automated pipeline for constructing emotion-centric statements, termed **INSETS** (**IN**telligent Vi**S**ual **E**motion **T**agger and **S**tatement Constructor). It operates through two stages: open-vocabulary emotion tagging and emotion statement construction, both of which build upon the well-established Parrott's Hierarchical emotion model (Parrott, 2001). This tree-structured taxonomy organizes emotions into 6 primary, 25 secondary, and 113 tertiary categories (Appendix D), where the primary level includes three positive emotions (joy, love, surprise) and three negative emotions (anger, fear, sadness). Secondary emotions elaborate these categories with greater diversity, while tertiary emotions refine them into more specific affective states.

## 4.1  OPEN-VOCABULARY EMOTION TAGGING

At this stage, INSETS aims to assign open-vocabulary emotion labels for images, laying a solid foundation for constructing meaningful emotion-centric statements, with its procedure depicted in Figure 2. According to (Cheng et al., 2024), MLLMs demonstrate promising capabilities in generating emotional descriptions from visual content and extracting underlying emotions from these

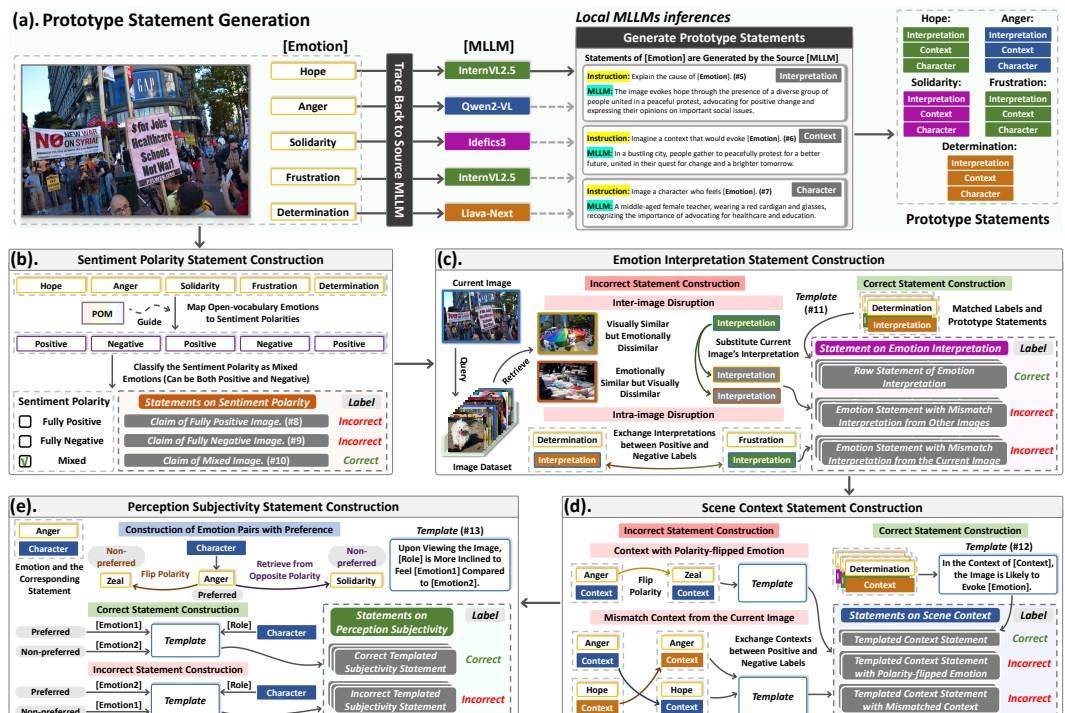

Figure 3: Illustration of the emotional statement construction stage. It begins with prototype statement generation (a) for each emotion label, which is distributed across multiple MLLMs. Then, based on the assigned emotion labels and the corresponding prototype statements, correct and incorrect emotion-centric statements are constructed from four dimensions: sentiment polarity (b), emotion interpretation (c), scene context (d), and perception subjectivity (e).

descriptions. However, challenges such as hallucinations (Bai et al., 2024), trustworthiness issues (Liu et al., 2024b), and inherent limitations in emotional perception can lead to inaccuracies in the extracted emotions. To enhance reliability, we devise an ensemble-based majority voting mechanism, aggregating outputs from multiple MLLMs to cross-validate and refine emotion label assignments.

Given an image sample, we first extract its potential open-vocabulary emotions from multiple MLLMs. MLLMs are prompted to analyze the emotions evoked by the image (with #1 prompt in Table 7, abbreviated as "#1" in the following) and then extract emotions applicable to the image (#2) [Figure 2 (a)]. This process is iteratively applied to all images in the dataset, aggregating potential emotions into an emotion pool. Next, we refine this pool by filtering out words unsuitable as emotion descriptors (#3), using GPT-4 (OpenAI, 2023) as the judge due to its superior linguistic emotional perception (Sabour et al., 2024) [b]. Once the filtered emotion pool is obtained, we attach the remaining emotions to Parrott's hierarchical emotion model [c]. GPT-4 is prompted to categorize each open-vocabulary emotion into the closest tertiary emotion in Parrott's model (#4), followed by manual refinement from a hired human expert. This process results in an extended version of Parrott's model, which we refer to as the **P**arrott-based **O**pen-vocabulary Hierarchical **M**odel (**POM**) [d]. This unified framework enables multi-level tracing of affective states for each open-vocabulary emotion, facilitating more accurate and interpretable emotion tagging.

Subsequently, leveraging POM, the ensemble-based majority voting mechanism selects consensus open-vocabulary emotion labels for images [e]. Specifically, emotions extracted from multiple MLLMs are first mapped to secondary categories, where model voting allocates quotas. Within each category, candidate labels are ranked by frequency, and the top-ranked ones are selected accordingly. This procedure enhances the reliability of annotations while retaining open-vocabulary flexibility.

## 4.2 EMOTIONAL STATEMENT CONSTRUCTION

Building upon the assigned emotion labels, we construct automatically-annotated emotion-centric statements, as illustrated in Figure 3. The pipeline initiates with prototype statement generation [a].

Table 1: Statistics of the MLLMs employed in INSETS. For each MLLM, we report the number of parameters, the average extracted emotions per image, the number selected as emotion labels, and the proportion of prototype statements it generates.

| MLLMs | #P (B) | Extracted Emotion | Selected Emotion | Generated Statement |
|---|---|---|---|---|
| LLaVa-1.6 (Liu et al., 2024a) | 7.6 | 8.3 | 2.4 | 9.8% |
| Mantis (Jiang et al., 2024) | 8.5 | 12.6 | 2.9 | 13.1% |
| mPLUG-Owl3 (Ye et al., 2024) | 8.1 | 9.2 | 2.7 | 11.2% |
| Idefics3 (Laurençon et al., 2024) | 8.5 | 10.0 | 2.9 | 12.5% |
| Phi-3.5-Vision (Abdin et al., 2024) | 4.1 | 9.9 | 2.8 | 11.7% |
| Qwen2-VL (Wang et al., 2024) | 8.3 | 8.8 | 2.7 | 10.9% |
| Llama-3.2-Vision (Dubey et al., 2024) | 10.7 | 7.2 | 2.3 | 9.3% |
| Molmo (Deitke et al., 2024) | 8.0 | 10.8 | 2.7 | 12.0% |
| InternVL2.5 (Chen et al., 2024b) | 8.3 | 8.5 | 2.3 | 9.5% |

Table 2: Statistics of INSETS-462k and MVEI.

| INSETS-462k | |
|---|---|
| Number of Images | 17,716 |
| Number of Statements | 462,369 |
| Emotion Labels Per Image | 4.9 |
| Distinct Emotion Labels | 751 |
| Statements Per Image | 26.1 |
| Average Length of Statements | 39.0 |
| **MVEI** | |
| Number of Images | 3,086 |
| Number of Statements | 3,086 |
| Emotion Labels Per Image | 5.2 |
| Distinct Emotion Labels | 424 |
| Statements Per Image | 1.0 |
| Average Length of Statements | 37.0 |

For each emotion label, we trace it back to the MLLM that extracts it, prompting the MLLM to generate three prototype statements: **1).** *prototype interpretation* of the emotion by inquiring about the cause of the emotion (#5); **2).** *prototype context* that aligns with the emotion by requesting a background story (#6); and **3).** *prototype character* who would experience the emotion by questioning the possible identity of the viewer (#7). From the dataset perspective, the prototype generation is distributed across multiple MLLMs, ensuring diversity in the subsequent statement construction.

***Sentiment Polarity Statement Construction*** [b]: We classify the sentiment polarity of each image into three mutually exclusive categories according to POM: **1).** *Fully Positive* when all labels reside in the positive spectrum; **2).** *Fully Negative* when all labels reside in the negative spectrum; **3).** *Mixed* when positive and negative labels both exist. Next, the ground truth correctness of three predefined statements on sentiment polarity (#8,9,10) is determined accordingly.

***Emotion Interpretation Statement Construction*** [c]: Each statement is constructed by combining a prototype interpretation with an emotional state (#11). Matched labels and prototype statements are assigned as correct, while mismatched ones are considered incorrect. We design two disruption strategies for each image: **1).** *Inter-image disruption* retrieves two images from the dataset—one exhibiting visual similarity but emotional dissimilarity to test whether MLLMs can comprehend the affective gap (Hanjalic, 2006), the other demonstrating emotional similarity but visual dissimilarity to evaluate whether MLLMs can identify the emotional triggers in images—and substitute the current prototype interpretation using one of theirs. Visual similarity is measured by CLIP-score (Radford et al., 2021), and emotional similarity is decided by tertiary emotions in POM. **2).** *Intra-image disruption* exchanges interpretations between labels of contrasting polarity within the same image, probing whether MLLMs can establish precise causal linkages between triggers and specific emotions.

***Scene Context Statement Construction*** [d]: Each statement is combined from a prototype context and an emotional conclusion (#12), where the construction of correct statements mirrors the previous case. For incorrect ones, we adopt two strategies: **1).** *a flip-polarity operation* that replaces the label with a tertiary emotion randomly sampled from the opposite spectrum in POM, and **2).** *swapping prototype contexts* between opposite-polarity labels within the same image.

***Perception Subjectivity Statement Construction*** [e]: We combine a prototype character with their inclination toward one of two candidate emotions (#13) to form a statement. For each character, the preferred emotion corresponds to its label, while the non-preferred emotion is obtained either from opposite-polarity labels within the same image or via flip-polarity sampling. Correct statements adopt the canonical preference order, whereas incorrect ones are formed by reversing it.

## 4.3 CONSTRUCTION OF INSETS-462K AND MVEI

Given the high quality of EmoSet (Yang et al., 2023a), we select 17,716 images from it as the image source for INSETS. We employ nine recent popular MLLMs with impressive performance (Contributors, 2023) for open-vocabulary emotion extraction and prototype statement generation. Their detailed participation is reported in Table 1. Observably, the final assigned emotion labels and prototype statements are evenly distributed across the MLLMs, ensuring diversity in the constructed data. In addition, a psychology postgraduate with formal training is hired to refine the attachment of

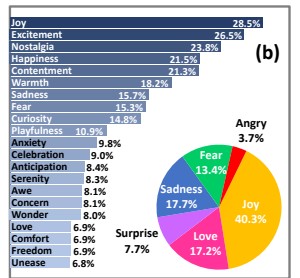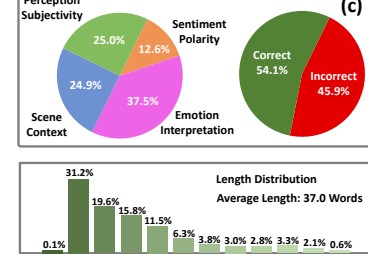

Figure 4: A closer gaze at MVEI. Illustrations of a sample (a), the distribution of emotion labels (b), and the distribution of emotion-centric statements (c).

open-vocabulary labels, which takes approximately 15 hours in total. Collectively, INSETS produces an automatically annotated ESJ corpus of 462K samples, namely INSETS-462k.

Based on this corpus, we sample 3,164 distinct image-statement pairs for careful human refinement. Five graduate students, each provided with detailed task instructions, are recruited to assess the accuracy of automatically assigned annotations. The statistics of human refinement are presented in Table 3, where annotators achieve consistently high agreement across the four task dimensions. For each pair, the annotation is deemed correct if at least four annotators reach consensus (5/5 or 4/5), incorrect if consensus is in the opposite direction (1/5 or 0/5), and ambiguous otherwise (3/5 or 2/5).

Table 3: Statistics of the human refinement process. *Kappa* represents *Fleiss' Kappa*.

| **MVEI** | Sentiment Polarity | Emotion Interpretation | Scene Context | Perception Subjectivity | Total |
|---|---|---|---|---|---|
| | Annotation Agreement (%) | | | | |
| 5/5 | 61.0 | 42.5 | 78.1 | 44.0 | 54.0 |
| 4/5 | 33.2 | 46.6 | 15.9 | 43.7 | 36.6 |
| 3/5 | 1.3 | 1.3 | 1.3 | 1.7 | 1.4 |
| 2/5 | 1.0 | 0.7 | 0.4 | 2.4 | 1.1 |
| 1/5 | 2.5 | 3.6 | 1.9 | 3.9 | 3.1 |
| 0/5 | 1.0 | 5.3 | 2.4 | 4.3 | 3.8 |
| *Kappa* | 0.68 | 0.51 | 0.81 | 0.52 | 0.61 |
| | Construction Accuracy (%) | | | | |
| ✓ Pairs | 94.9 | 86.2 | 94.6 | 87.5 | 89.7 |
| ✗ Pairs | 93.4 | 92.0 | 93.4 | 88.0 | 91.5 |

Overall, **90.6%** of the automated annotations are judged accurate—**89.7%** for correct statements and **91.5%** for incorrect statements—validating the high reliability of INSETS. After retaining correct labels, rectifying errors, and discarding ambiguous cases, we derive the final **MVEI** benchmark (**M**ultifaceted evaluation of **V**isual **E**motion **I**ntelligence). MVEI comprises 3,086 samples with over 400 distinct emotion labels, with detailed statistics provided in Table 2. Benefiting from the large-scale automatic construction of INSETS-462k, MVEI is far more labor-efficient than prior emotion evaluation benchmarks, requiring only about **100 person-hours** for the subsequent refinement.

## 5 ANALYSIS AND EVALUATION

### 5.1 DETAILS OF MVEI

To further characterize MVEI, we provide its fine-grained statistics in Figure 4. A sample is shown in Figure 4 (a), which includes five emotion labels and an emotion-centric statement. Figure 4 (b) illustrates the distribution of popular emotion labels, where the most frequent labels include Joy, Excitement, Nostalgia, Happiness, and Contentment. When mapped to the primary emotions in POM, Joy dominates (40.3%), followed by Sadness (17.7%), Love (17.2%), Fear (13.4%), Surprise (7.7%), and Anger (3.7%). This distribution reflects broad coverage of affective states. Finally, Figure 4 (c) presents statistics of the statements, showing a natural length distribution and a balanced spread across the four evaluation dimensions as well as correct/incorrect labels.

### 5.2 EVALUATION PREPARATIONS

To evaluate MLLMs with ESJ, each model is given an image–statement pair and prompted to judge its correctness. The prompt is formulated as: *"Based on the provided image and emotional statement, please determine whether the statement aligns with the content of the image. If it does, respond with*

Table 4: Evaluation of popular MLLMs on MVEI. For fair comparison, we separate the MLLMs involved in constructing INSETS-462k (the upper part) and the others (the lower part). The highest values in each section are marked in **bold**, underline, and wavy underline, respectively.

| MLLMs | #Param | Accuracy (%) | | | | | Positive Ratio | Give-up Ratio |
|---|---|---|---|---|---|---|---|---|
| | | Sentiment Polarity | Emotion Interpretation | Scene Context | Perception Subjectivity | Total | | |
| LLaVa-1.6 (Liu et al., 2024a) | 7.6B | 66.4 | 69.7 | 55.3 | 49.7 | 60.2 | 18.4 | 0 |
| Mantis (Jiang et al., 2024) | 8.5B | 61.2 | 65.9 | 67.2 | 61.2 | 64.4 | 84.4 | 0.1 |
| mPLUG-Owl3 (Ye et al., 2024) | 8.1B | 73.9 | 79.3 | 81.7 | 75.0 | 78.1 | 67.3 | 0 |
| Idefics3 (Laurençon et al., 2024) | 8.5B | 75.4 | 78.6 | 75.5 | 62.6 | 73.4 | 49.5 | 0.2 |
| Phi-3.5-Vision (Abdin et al., 2024) | 4.1B | 74.7 | 72.5 | 82.6 | 74.8 | 75.9 | 64.1 | 0 |
| Qwen2-VL (Wang et al., 2024) | 8.3B | 70.7 | 75.0 | 86.1 | 72.8 | 76.6 | 65.7 | 0 |
| Llama-3.2-Vision (Dubey et al., 2024) | 10.7B | 68.7 | 75.9 | 85.2 | 72.0 | 76.3 | 71.2 | 0.2 |
| Molmo (Deitke et al., 2024) | 8.0B | 61.4 | 76.0 | 79.2 | 59.4 | 70.7 | 38.1 | 0 |
| InternVL2.5 (Chen et al., 2024b) | 8.3B | 75.7 | 80.2 | 79.4 | 61.3 | 74.7 | 52.9 | 0.2 |
| BLIP2 (Li et al., 2023c) | 7.7B | 51.1 | 52.8 | 55.4 | 52.5 | 53.2 | 96.8 | 2.5 |
| InstructBLIP (Dai et al., 2023) | 7.9B | 29.8 | 40.5 | 33.9 | 37.8 | 36.8 | 43.8 | 37.5 |
| Otter (Li et al., 2023a) | 8.2B | 32.6 | 21.4 | 32.1 | 27.2 | 27.0 | 9.9 | 52.1 |
| DeepSeek-VL (Lu et al., 2024) | 7.3B | 68.7 | 70.8 | 81.1 | 73.2 | 73.7 | 73.1 | 0 |
| Paligemma (Beyer et al., 2024) | 2.9B | 50.6 | 46.3 | 49.3 | 45.7 | 47.4 | 49.4 | 5.5 |
| MiniCPM (Yao et al., 2024) | 8.7B | 70.4 | 78.4 | 81.9 | 70.5 | 76.2 | 66.0 | 0 |
| Qwen2.5-VL (Team, 2025) | 8.3B | 63.2 | 81.5 | 83.9 | 66.3 | 75.9 | 45.9 | 0 |
| GPT4o-mini (Hurst et al., 2024) | – | 62.5 | 80.0 | 78.9 | 71.8 | 75.4 | 49.5 | 0 |
| GPT4o (Hurst et al., 2024) | – | 72.5 | 84.3 | 81.6 | 69.2 | 78.3 | 65.0 | 1.6 |

*Correct*. *If it does not, respond with **Incorrect**."* Each image-statement pair is queried three times per model, and the most frequent response is selected as the final decision. Accuracy serves as the primary evaluation metric. As identified in prior work (Li et al., 2023d), some MLLMs may exhibit a strong bias toward either positive or negative responses, which may compromise accuracy-based evaluation validity. To address this, we introduce two diagnostic metrics: *Positive Ratio* calculates the proportion of "Correct" among all responses; *Give-up Ratio* measures the proportion of cases where the MLLM fails to provide either judgment.

We evaluate a wide range of MLLMs on MVEI, including both open-source and closed-source ones. Besides the MLLMs employed in constructing INSETS-462k, we also incorporate: BLIP-2 (Li et al., 2023c), InstructBLIP (Dai et al., 2023), Otter (Li et al., 2023a), Deepseek-VL (Lu et al., 2024), Paligemma (Beyer et al., 2024), MiniCPM (Yao et al., 2024), Qwen2.5-VL (Team, 2025), GPT4o-mini, and GPT4o (Hurst et al., 2024).

### 5.3 RESULTS AND FINDINGS

***Comparison of MLLMs (Table 4)***: Overall, recent MLLMs substantially outperform earlier ones, which often suffer from severe response biases or instruction-following failures. These results suggest that advancements in general visual tasks also benefit emotional perception. Among state-of-the-art MLLMs, their capabilities vary noticeably across different task dimensions, with no single MLLM achieving top performance in all categories. For instance, InternVL2.5 and GPT4o excel at recognizing basic emotional tones and performing affective reasoning, yet exhibit relative shortcomings in contextual and subjective emotion prediction. These results highlight the multifaceted challenges of visual emotion understanding and the need for continued targeted development.

***Comparison with Human Performance (Table 5)***: We evaluate 25 human participants alongside leading MLLMs on a 300-sample subset of MVEI. The results show that humans achieve an average overall accuracy of 91.6%, substantially surpassing both open-source and proprietary MLLMs. The performance gap is most evident in determining sentiment polarity and understanding perception subjectivity. Given that MLLMs perform comparatively well in emotion interpretation, their limitations in polarity appear to stem from an overreliance on provided affective cues and difficulty in distinguishing boundaries between sentiment categories. In the case of perception subjectivity, the gap seems more fundamental, reflecting MLLMs' limited proficiency to capture individual differences in emotional perception. Collectively, these findings suggest that current MLLMs **may not yet be sufficiently competent for LLM-as-a-judge applications in affective perception tasks**. They underscore the need for more rigorous benchmarking of foundational capabilities, while also pointing to considerable potential for future advancement.

Table 5: Evaluation of humans on a 300-sample subset of MVEI. To ensure fairness, the results of partial leading MLLMs are also reported, adhering to the same partition as Table 4.

| MLLMs | #Param | Accuracy (%) | | | | | Positive Ratio | Give-up Ratio |
| --- | --- | --- | --- | --- | --- | --- | --- | --- |
| | | Sentiment Polarity | Emotion Interpretation | Scene Context | Perception Subjectivity | Total | | |
| mPLUG-Owl3 (Ye et al., 2024) | 8.1B | 74.6 | **80.4** | 82.9 | **77.2** | **79.5** | 67.3 | 0 |
| Phi-3.5-Vision (Abdin et al., 2024) | 4.1B | 75.4 | 72.9 | **83.9** | 73.3 | 76.1 | 64.0 | 0 |
| InternVL2.5 (Chen et al., 2024b) | 8.1B | **77.2** | 79.5 | 79.3 | 63.2 | 75.1 | 52.1 | 0 |
| DeepSeek-VL (Lu et al., 2024) | 7.3B | 70.2 | 70.5 | 80.2 | **73.7** | 73.7 | 73.5 | 0 |
| MiniCPM (Yao et al., 2024) | 8.7B | 70.2 | 78.9 | 82.4 | 72.4 | 77.1 | 65.4 | 0 |
| Qwen2.5-VL (Team, 2025) | 8.3B | 64.0 | 81.5 | **83.3** | 68.0 | 76.4 | 47.4 | 0 |
| GPT4o-mini (OpenAI, 2023) | - | 64.0 | 79.2 | 77.5 | 71.3 | 74.9 | 49.8 | 0 |
| GPT4o (OpenAI, 2023) | - | **73.7** | **84.5** | 81.2 | 71.1 | **79.0** | 64.6 | 0.6 |
| Human Average | – | 92.3 | 90.1 | 95.3 | 89.6 | 91.6 | 53.4 | 0 |
| Human Best | – | 97.4 | 95.8 | 98.7 | 94.7 | 95.2 | – | – |

Table 6: Evaluation of lightweight MLLMs adaptation techniques on MVEI.

| Qwen2.5-VL (8.3B) | #Shot | Accuracy (%) | | | | |
| --- | --- | --- | --- | --- | --- | --- |
| | | Sentiment Polarity | Emotion Interpretation | Scene Context | Perception Subjectivity | Total |
| Direct Inference | - | 63.2 | 81.5 | 83.9 | 66.3 | 75.9 |
| Chain-of-Thought Reasoning | - | 67.4 (+4.2) | 81.5 (+0.0) | 84.6 (+0.7) | 67.0 (+0.7) | 76.6 (+0.8) |
| In-Context Learning: Random Retrieval | 2 | 66.3 (+3.1) | 81.6 (+0.1) | 84.8 (+0.9) | 66.5 (+0.2) | 76.9 (+1.0) |
| In-Context Learning: Random Retrieval | 4 | 68.8 (+5.6) | 81.7 (+0.2) | 85.0 (+1.1) | 66.7 (+0.4) | 77.1 (+1.2) |
| In-Context Learning: Random Retrieval | 8 | 70.1 (+6.9) | 81.7 (+0.2) | 84.9 (+1.0) | 67.0 (+0.7) | 77.3 (+1.4) |
| LoRA Fine-Tuning (Hu et al., 2022) | - | 78.6 (+15.4) | 84.7 (+3.2) | 86.3 (+2.4) | 70.3 (+4.0) | 80.7 (+4.8) |
| Full Parameter Fine-Tuning (Freeze Vision) | - | 84.3 (+21.1) | 84.8 (+3.3) | 87.0 (+3.1) | 71.1 (+4.8) | 81.9 (+6.0) |
| GRPO (Shao et al., 2024) | - | 83.2 (+20.0) | 82.5 (+1.0) | 86.5 (+2.6) | 71.1 (+4.8) | 80.7 (+4.8) |

***Influence of MLLM adaptations (Table 6)***: To delve deeper into MLLMs' emotional intelligence and shed light on the potential influence of model adaptation, we adapt Qwen2.5-VL using several popular techniques and evaluate their impact on MVEI. For in-context learning, demonstrations are randomly retrieved from the corresponding task dimensions of INSETS-462k, with overlapping MVEI samples excluded. For parameter-efficient fine-tuning, we apply LoRA (Hu et al., 2022) on a 10k-sample subset of INSETS-462k (excluding MVEI overlaps), using a learning rate of 1e-5 and LoRA rank of 16. The full-parameter fine-tuning is conducted on the same subset with an identical learning rate, during which the vision encoder is frozen. Finally, for GRPO (Shao et al., 2024), we train on the same subset with a learning rate of 1e-6 and perform 4 rollouts per query.

As shown in Table 6, all applied techniques consistently improve MLLM performance, demonstrating the benefits of both in-context learning and task-specific fine-tuning. The most pronounced gains occur in sentiment polarity, indicating that MLLMs possess the capability to capture overall emotional tone. Their previous deficiency is likely due to confusion between positive, negative, and mixed categories, and it can be effectively alleviated through few-shot demonstrations or lightweight fine-tuning. By contrast, perception subjectivity shows only modest improvement and remains the weakest dimension, reflecting a more fundamental challenge that may require subjectivity-oriented pre-training objectives or specialized datasets. While targeted adaptation on INSETS-462k provides clear benefits, we view this work primarily as a foundational benchmark for advancing emotional intelligence in more general-purpose MLLMs. Rather than treating ESJ as a direct optimization target, we advocate its use as an evaluation metric and feedback signal to guide broader model development.

## 6 CONCLUSION

In this paper, we introduce the Emotion Statement Judgment task and the INSETS pipeline, which jointly address the incompatibility of conventional emotion evaluation approaches with MLLMs. Building on these components, we construct the MVEI benchmark and the large-scale INSETS-462K corpus in a labor-efficient manner, aiming to advance open-vocabulary, multifaceted, and scalable visual emotion evaluation in MLLMs. Grounded in psychological theory, MVEI evaluates four complementary dimensions of affective cognition: sentiment polarity, emotion interpretation, scene context, and perception subjectivity. Comprehensive experiments on MVEI reveal that, while

current MLLMs demonstrate certain competence in interpreting basic emotions, contextual cues, and the associations between triggers and affective states, they still fall substantially short of human performance. In particular, their limitations are most evident in handling perception subjectivity, which remains a fundamental challenge even after targeted model adaptations. Taken together, this work establishes a foundation for advancing the study of emotional intelligence in MLLMs, aiming to foster future research in both MLLM development and AICA.

## 7 ACKNOWLEDGEMENT

This work is supported by the National Natural Science Foundation of China (Grant No. 62406318 & 62376266 & 62571294 & 62441614) and the Beijing Natural Science Foundation (L252009).

## 8 ETHICS STATEMENT

This study evaluates MLLMs in visual emotion comprehension through a new task, pipeline, and benchmark. While we aim to advance research, several ethical considerations merit attention.

**First, the dataset exhibits certain distributional imbalances in the emotion label of images**. The images used in this work originate from EmoSet, which ultimately traces back to user-generated posts on social media. Such posts naturally reflect platform-specific emotional biases, most notably, positive content typically appears more frequently than negative content (Niu et al., 2016). This characteristic carries over to our MVEI benchmark, where, as shown in Figure 4, positive-polarity images account for 65.2% of the data, compared to 34.8% for negative ones. While this skew partially mirrors real-world content distributions, it may influence model behavior or downstream analysis if not interpreted carefully. We therefore encourage users to remain aware of these imbalances to avoid biased or misleading conclusions.

**Second, perception subjectivity statements may contain latent demographic biases arising from the automatically generated characters.** Although extreme or inappropriate cases are filtered through human refinement, demographic attributes, such as age, gender, or cultural background, may still be unevenly reflected or stereotypically implied by the MLLMs. And since the characters are produced without explicit structural control, systematically quantifying their demographic statistics remains challenging. This limitation introduces the risk of subtle demographic skew being propagated or reinforced through the benchmark. Empirical quantification of these demographic patterns would enable finer-grained evaluation and customization, and we regard this as an important direction for future development.

**Third, MLLM-generated data may still exhibit cultural-perspective and aesthetic-perception biases.** Prior work has shown that cultural tendencies embedded in training corpora, such as language distribution and region-specific viewpoints, can be amplified during LLM inference, and even multilingual models often fail to equitably represent diverse cultural values (Tao et al., 2024). In addition, LLMs have been found to exhibit aesthetic preferences that may implicitly reinforce stereotypical standards (Kotek et al., 2023). Such tendencies could be potentially carried over into the constructed emotional labels and statements. Although the bias of any specific model can be alleviated through the proposed ensemble-based majority voting mechanism, these biases cannot be fully eliminated. Users should therefore interpret culturally sensitive results with caution and avoid overgeneralizing findings.

**Fourth, annotation-level biases due to cultural differences or personal experiences may persist.** Emotion perception is inherently subjective, and these differences can shape annotation outcomes. While ESJ task formulation targets specifically for such issues, it can hardly guarantee the complete elimination of these biases. Therefore, these concerns also warrant caution from users.

Although addressing these ethical concerns falls beyond the immediate scope of this study, we document them here to maintain transparency. We hope this clarifies the limitations of the benchmark and supports future efforts toward mitigation. All data used in this work are drawn from publicly available benchmarks, and no private information was collected or disclosed. By acknowledging these considerations, we aim to promote responsible use of our data, mitigate potential risks, and support its positive impact on future research.

## 9 REPRODUCIBILITY STATEMENT

To ensure reproducibility, the manuscript provides comprehensive documentation of the INSETS implementation, the human refinement process, and detailed statistics of both the INSETS-462K corpus and the MVEI benchmark. We release code and data on: https://github.com/wdqqdw/MVEI.

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

## A   LIMITATIONS

Several limitations in this work can be further improved. *First*, our evaluation primarily focuses on MLLMs with parameters under 10B due to computational constraints imposed by hardware. Although this covers practical deployment scenarios, it excludes larger-scale open-source MLLMs that may exhibit superior visual emotion perception capabilities. *Second*, the current implementation is limited to monolingual evaluation. Yet we highlight that adapting INSETS for multilingual construction would require relatively limited engineering effort, primarily involving adjustments in MLLM selection, prompt design, and template configuration. Moreover, while we explored lightweight model adaptations, more nuanced or advanced strategies remained underexplored.

## B   LLM USAGE

This paper employs (M)LLMs for prompt engineering and data annotation. Additionally, they are also used during manuscript writing, mainly for grammar checking and refinement.

## C   PROMPTS AND STATEMENT TEMPLATES

The prompts and statement templates used in the INSETS pipeline are presented in Table 7.

Table 7: Prompts and statement templates employed in the INSETS pipeline.

|     | Prompts and Statement Templates |
| --- | --- |
| #1 | You are an Emotional Perception Expert. Please analyze the emotions that might be evoked by the given image. Your analysis should explore a wide range of visual attributes, such as brightness, colorfulness, depicted scenes, objects, human actions, and facial expressions. Additionally, provide detailed explanations linking these attributes to the emotions they may trigger. If applicable, discuss any potential cultural or psychological factors influencing these emotional responses. |
| #2 | You are an Emotional Perception Expert. Your task is to extract all applicable emotions as comprehensively as possible based on the image description. Focus on distinct emotions such as happiness, sadness, fear, anger, etc. Keep the list concise, with a maximum of 10 distinct emotions. |
| #3 | You are tasked with determining whether the word "[**word**]" describes a specific emotional state. An emotional state is a psychological condition involving feelings and reactions triggered by internal or external events. Respond with "Yes" if the word aligns with this definition, or "No" otherwise. The output format should be {"word": "response"}. |
| #4 | You are tasked with assigning the word "[**word**]" to the most closely related emotional category from the following 115 predefined options: "[**categories**]". Consider broader semantic connections and possible emotional nuances when making your judgment. If the word cannot reasonably fit any category, respond with "not applicable". Do not create or assign new categories outside of the provided list. Do not provide any explanations or reasons for your choice. The output format should be {"word": "response"}. |
| #5 | Briefly explain why this image might evoke "[**emotion**]" in viewers, without mentioning any other emotions. |
| #6 | Imagine a background story for the image that would evoke a sense of "[**emotion**]" in viewers. Respond in one sentence. Do not mention the content in the image. |
| #7 | Imagine a character who would feel "[**emotion**]" when viewing this image. Include details such as their age, gender, profession, and other relevant traits. Describe the character in one concise sentence without further explanation. |
| #8 | Upon viewing this image, observers, despite various individual or contextual factors, are most likely to experience positive emotions. |
| #9 | Upon viewing this image, observers, despite various individual or contextual factors, are most likely to experience negative emotions. |
| #10 | Upon viewing this image, observers are equally likely to experience either positive or negative emotions, depending on individual or contextual factors. |
| #11 | Therefore, the image might evoke "[**emotion**]" in viewers. |
| #12 | In the context of: "[**context**]", the image is likely to evoke a sense of "[**emotion**]". |
| #13 | Upon viewing the image, "[**role**]" is more inclined to feel "[**emotion1**]" compared to "[**emotion2**]". |

## D   DETAILS OF PARROTT'S HIERARCHICAL MODEL

We present the complete emotion taxonomy of Parrott's hierarchical model in Table 8.

Table 8: Emotion taxonomy of Parrott's hierarchical model.

| Primary Emotion | Secondary Emotion | Tertiary Emotion |
|---|---|---|
| Love | Affection | Adoration, Fondness, Liking, Attraction, Caring, Tenderness, Compassion, Sentimentality |
| | Lust | Desire, Passion, Infatuation |
| | Longing | Longing |
| Joy | Cheerfulness | Amusement, Bliss, Gaiety, Glee, Jolliness, Joviality, Joy, Delight, Enjoyment, Gladness, Happiness, Jubilation, Elation, Satisfaction, Ecstasy, Euphoria |
| | Zest | Enthusiasm, Zeal, Excitement, Thrill, Exhilaration |
| | Contentment | Pleasure |
| | Pride | Triumph |
| | Optimism | Eagerness, Hope |
| | Enthrallment | Enthrallment, Rapture |
| | Relief | Relief |
| Surprise | Surprise | Amazement, Astonishment |
| Anger | Irritability | Aggravation, Agitation, Annoyance, Grouchy, Grumpy, Crosspatch |
| | Exasperation | Frustration |
| | Rage | Anger, Outrage, Fury, Wrath, Hostility, Ferocity, Bitterness, Hatred, Scorn, Spite, Vengefulness, Dislike, Resentment |
| | Disgust | Revulsion, Contempt, Loathing |
| | Envy | Jealousy |
| | Torment | Torment |
| Sadness | Suffering | Agony, Anguish, Hurt |
| | Sadness | Depression, Despair, Gloom, Glumness, Unhappiness, Grief, Sorrow, Woe, Misery, Melancholy |
| | Disappointment | Dismay, Displeasure |
| | Shame | Guilt, Regret, Remorse |
| | Neglect | Alienation, Defeatism, Dejection, Embarrassment, Homesickness, Humiliation, Insecurity, Insult, Isolation, Loneliness, Rejection |
| | Sympathy | Pity, Mono no aware, Sympathy |
| Fear | Horror | Alarm, Shock, Fear, Fright, Horror, Terror, Panic, Hysteria, Mortification |
| | Nervousness | Anxiety, Suspense, Uneasiness, Apprehension, Worry, Distress, Dread |

## E  FORMALIZATION OF THE MAJORITY-VOTING MECHANISM

In this section, we provide a formalized definition of the majority-voting mechanism for clarification and transparency. Let an image be processed by $n$ MLLMs. The set of open-vocabulary emotion labels generated by the $i$-th model is denoted by $L_i = \{e_{i,1}, e_{i,2}, \ldots, e_{i,m}\}$, where $i \in \{1, 2, \ldots, n\}$.

Let $\mathcal{P}$ be the mapping from an open-vocabulary emotion to the secondary emotion in POM. The ensemble-based majority-voting procedure is defined as follows.

1. **Secondary-Emotion Quota.** For a secondary emotion $\bar{e}$, define its quota $Q(\bar{e})$ as:

$$Q(\bar{e}) = \max\left( \left\lfloor \sum_{i=1}^{n} \mathbb{I}[\bar{e} \in \{\mathcal{P}(e) \mid e \in L_i\}] - \frac{n}{2} + 1 \right\rfloor, 0 \right)$$

where $\mathbb{I}[\cdot]$ is the indicator function. This quantity is strictly positive only when $\bar{e}$ is supported by a majority of the models.

2. **Candidate Pool Formation.** The candidate pool of open-vocabulary labels for the secondary emotion $\bar{e}$ is defined as:

$$S(\bar{e}) = \left\{ e_{i,j} \mid i \in \{1, 2, \ldots, n\}; \ j \in \{1, 2, \ldots, m\}; \ \mathcal{P}(e_{i,j}) = \bar{e} \right\}.$$

3. **Consensus Selection.** Let $\mathrm{freq}(e)$ denotes the frequency of an open-vocabulary label $e$ in $S(\bar{e})$: $\mathrm{freq}(e) = |\{e_{i,j} \in S(\bar{e}) | e_{i,j} = e\}|$. The consensus labels for $\bar{e}$ are defined as the top-$Q(\bar{e})$ unique labels in $S(\bar{e})$ ranked by $\mathrm{freq}(\cdot)$, where ties are resolved uniformly at random. The final consensus label set for the image is obtained by taking the union over all secondary emotions:

$$L_{\mathrm{cons}} = \bigcup_{\bar{e}} \mathrm{Top}_{Q(\bar{e})}(S(\bar{e})).$$

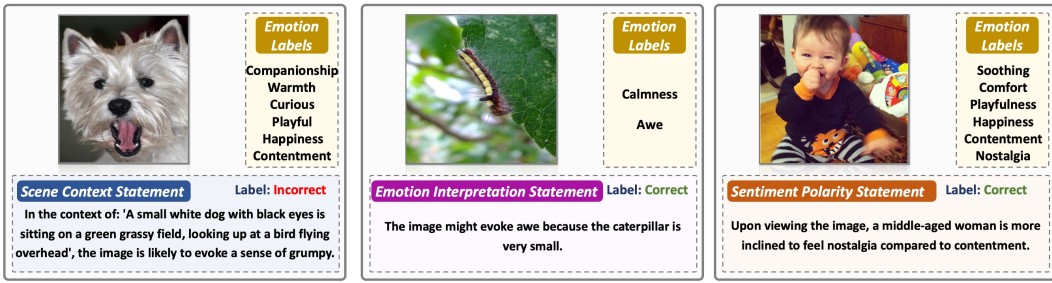

Figure 5: Samples that are deemed ambiguous during the human refinement process.

## F    VISUALIZATION OF AMBIGUOUS SAMPLES

Figure 5 presents three representative samples that are identified as ambiguous during the manual refinement process and thus excluded from the MVEI benchmark. *In the left case*, the contextual description is considered an inappropriate or overextended inference from the visual content. *In the middle case*, the emotional interpretation is overly brief, with cues that were too general or vague to support a reliable emotional conclusion. *In the right case*, the characterization lacked sufficient specificity, where two distinct emotions remain equally plausible.

We attribute these ambiguities to two principal factors: the inherent limitations of MLLMs in visual perception, which manifest as inaccurate descriptions or superficial analyses, and the intrinsic challenges of emotion-related data construction, where contextual subjectivity can render multiple interpretations valid. These observations collectively highlight the critical necessity of human refinement in data construction pipelines involving MLLMs.

## G    AGREEMENT BETWEEN ASSIGNED LABELS AND EMOSET LABELS

To further evaluate the automatically assigned open-vocabulary labels, we perform a cross-validation based on the original EmoSet labels. Since EmoSet is built upon the Mikels model (Mikels et al., 2005), which contains eight emotion categories (amusement, awe, contentment, excitement, anger, disgust, fear, and sadness), it is not naturally aligned with our adopted Parrott hierarchical model.

However, we note that except for awe, the remaining seven Mikels categories are distributed across different levels of Parrott's model. To enable comparison, we map them to the primary-level emotions in Parrott's model: amusement → joy, contentment → joy, excitement → joy, anger → anger, disgust → anger, fear → fear, sadness → sadness. Finally, using the open-vocabulary attachment obtained from our constructed POM, we map awe → surprise. Based on this mapping, our analysis shows that the automatically assigned open-vocabulary labels of 97.3% of the 17,716 images in INSETS-462k overlap with their original EmoSet labels at the primary level of the Parrott model. This high level of consistency provides strong evidence for the validity of our automated annotation pipeline.

## H    ADDITIONAL RESULTS OF MLLMS ADAPTATION TECHNIQUES

Table 9 reports results of more comprehensive MLLM adaptation techniques on MVEI, building upon those in Table 6. The 459k samples used for supervised fine-tuning correspond to the full INSETS-462k dataset with MVEI excluded, and the 50k samples used for reinforcement learning are a sampled subset of the former. For inference and GRPO, the CoT prompt are constructed by appending "think step-by-step" to the original prompt.

Table 9: Comprehensive Evaluation of MLLMs adaptation techniques on MVEI.

| Qwen2.5-VL (8.3B) | #Shot | #Sample | Accuracy (%) | | | | |
|---|---|---|---|---|---|---|---|
| | | | Sentiment Polarity | Emotion Interpretation | Scene Context | Perception Subjectivity | Total |
| Direct Inference | - | - | 63.2 | 81.5 | 83.9 | 66.3 | 75.9 |
| Chain-of-Thought (CoT) Reasoning | - | - | 67.4 | 81.5 | 84.6 | 67.0 | 76.6 |
| In-Context Learning: Random Retrieval | 2 | - | 66.3 | 81.6 | 84.8 | 66.5 | 76.6 |
| In-Context Learning: Random Retrieval | 4 | - | 68.8 | 81.7 | 85.0 | 66.7 | 77.1 |
| In-Context Learning: Random Retrieval | 8 | - | 70.1 | 81.7 | 84.9 | 67.0 | 77.3 |
| LoRA Fine-Tuning (Hu et al., 2022) | - | 10k | 78.6 | 84.7 | 86.3 | 70.3 | 80.7 |
| LoRA Fine-Tuning (Hu et al., 2022) | - | 459k | 82.2 | 86.0 | 86.9 | 71.9 | 82.2 |
| Full Parameter Fine-Tuning (Freeze Vision) | - | 10k | 84.3 | 84.8 | 87.0 | 71.1 | 81.9 |
| Full Parameter Fine-Tuning (Freeze Vision) | - | 459k | 85.6 | 86.5 | 87.6 | 73.3 | 83.4 |
| GRPO (Shao et al., 2024) (without CoT) | - | 10k | 83.2 | 82.5 | 86.5 | 71.1 | 80.7 |
| GRPO (Shao et al., 2024) (without CoT) | - | 50k | 84.0 | 82.7 | 86.3 | 71.4 | 80.9 |
| GRPO (Shao et al., 2024) (with CoT) | - | 10k | 86.2 | 82.9 | 86.6 | 72.3 | 81.6 |
| GRPO (Shao et al., 2024) (with CoT) | - | 50k | 86.8 | 83.0 | 87.2 | 72.7 | 82.0 |

# I    LINKS OF MLLMS

We provide the links to the model cards of the MLLMs we evaluated in the experiments.

LLaVa-1.6 (Liu et al., 2024a)

https://huggingface.co/llava-hf/llava-v1.6-mistral-7b-hf

Mantis (Jiang et al., 2024)

https://huggingface.co/TIGER-Lab/Mantis-8B-siglip-llama3

mPLUG-Owl3 (Ye et al., 2024)

https://huggingface.co/mPLUG/mPLUG-Owl3-7B-241101

Idefics3 (Laurençon et al., 2024)

https://huggingface.co/HuggingFaceM4/Idefics3-8B-Llama3

Phi-3.5-Vision (Abdin et al., 2024)

https://huggingface.co/microsoft/Phi-3.5-vision-instruct

Qwen2-VL (Wang et al., 2024)

https://huggingface.co/Qwen/Qwen2-VL-7B-Instruct

Llama-3.2-Vision (Dubey et al., 2024)

https://huggingface.co/meta-llama/Llama-3.2-11B-Vision-Instruct

Molmo (Deitke et al., 2024)

https://huggingface.co/allenai/Molmo-7B-D-0924

InternVL2.5 (Chen et al., 2024b)

https://huggingface.co/OpenGVLab/InternVL2_5-8B

BLIP2 (Li et al., 2023c)

https://huggingface.co/Salesforce/blip2-opt-6.7b-coco

InstructBLIP (Dai et al., 2023)

https://huggingface.co/Salesforce/instructblip-vicuna-7b

Otter (Li et al., 2023a)

https://huggingface.co/luodian/OTTER-Image-LLaMA7B-LA-InContext

DeepSeek-VL (Lu et al., 2024)

https://huggingface.co/deepseek-ai/deepseek-vl-7b-chat

Paligemma (Beyer et al., 2024)

https://huggingface.co/google/paligemma-3b-pt-448

MiniCPM (Yao et al., 2024)

https://huggingface.co/openbmb/MiniCPM-o-2_6

Qwen2.5-VL (Team, 2025)

https://huggingface.co/Qwen/Qwen2.5-VL-7B-Instruct

## J VISUALIZATION OF MVEI

More samples from MVEI are visualized in Figure 6, Figure 7, Figure 8, Figure 9, Figure 10, Figure 11, Figure 12, Figure 13.

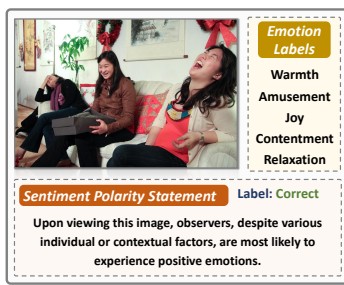 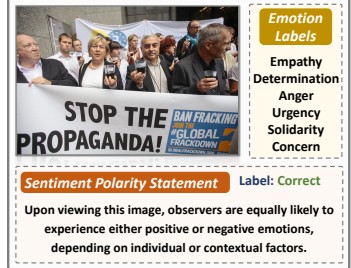 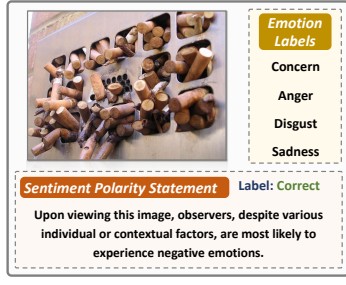

Figure 6: Sentiment polarity statements labeled as correct.

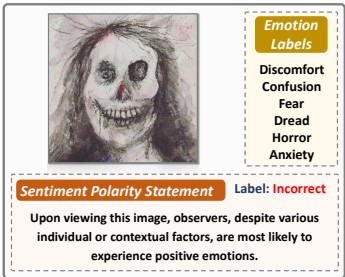 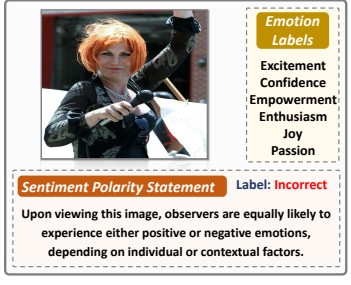 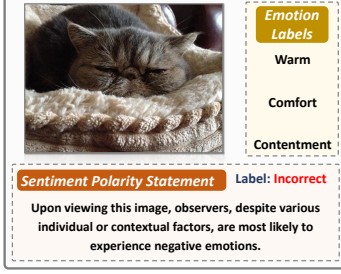

Figure 7: Sentiment polarity statements labeled as incorrect.

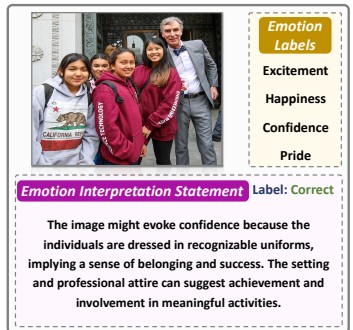 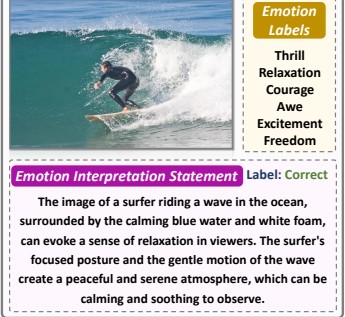 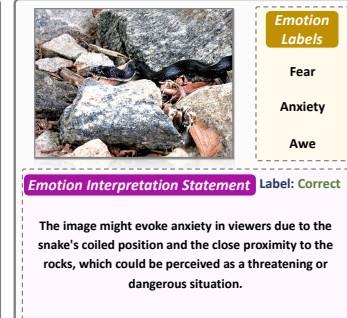

Figure 8: Emotion interpretation statements labeled as correct.

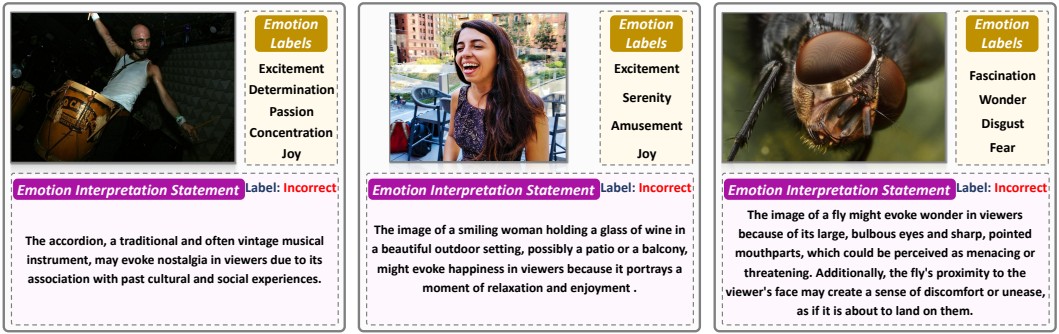

Figure 9: Emotion interpretation statements labeled as incorrect.

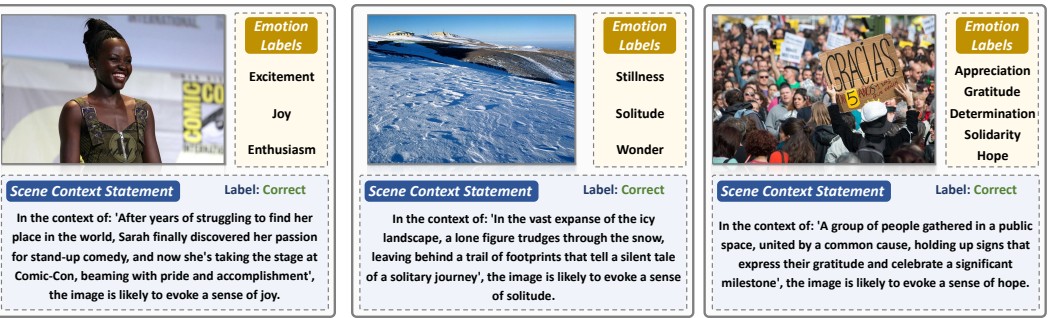

Figure 10: Scene context statements labeled as correct.

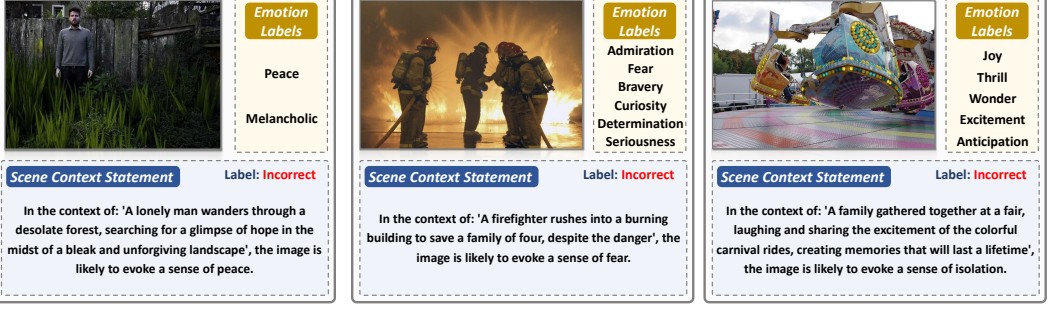

Figure 11: Scene context statements labeled as incorrect.

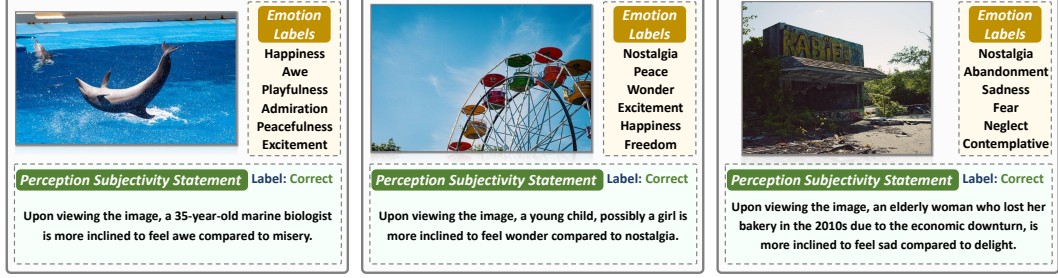

Figure 12: Perception subjectivity statements labeled as correct.

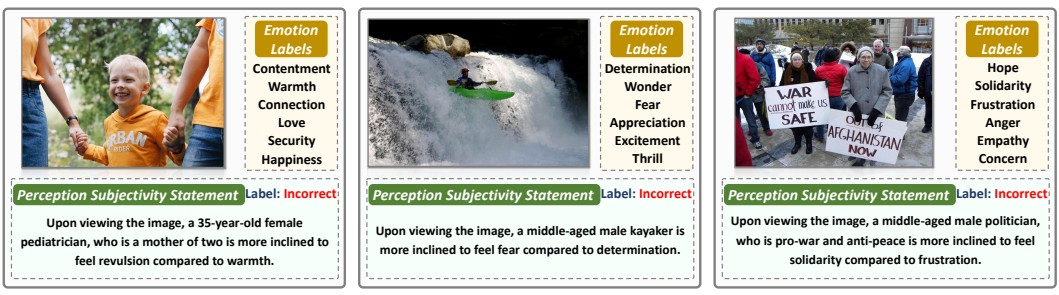

Figure 13: Perception subjectivity statements labeled as incorrect.

