# OpenReview forum: "Customizing Visual Emotion Evaluation for MLLMs: An Open-vocabulary, Multifaceted, and Scalable Approach"
_ICLR.cc/2026/Conference — ICLR 2026 Poster_

### Official Review · Reviewer_ck1T · 2025-10-24

**Soundness:** 3
**Presentation:** 3
**Contribution:** 3
**Rating:** 4
**Confidence:** 4

**Summary:**

The paper introduces a new annotation framework for MLLMs in the task of visual emotion estimation. The paper identifies limitations in the evaluation of MLLMs (namely, inaccurate responses, limited taxonomies, a focus on intrinsic image attributes, and reliance on majority voting) and first introduces a new task to address these limitations. Then, an automated annotation pipeline leveraging an ensemble of MLLMs is used to annotate 400k images and create fine-grained annotations that are used to produce statements with GPT4 and curated by a human expert. The labels and statements are further processed to produce additional coarse annotations (in Parrott-based hierarchy and positive/negative polarity). Finally, several popular MLLMs are benchmarked on the test set of the dataset, including an in-context learning and a finetuned version of Qwen.

**Strengths:**

- The problems identified by the paper are valid. The uncertain nature of human emotions (invoked, perceived or otherwise) is one of the key limitations in achieving high accuracy in the task, opposed to object detection, for example. Cultural barriers, inconsistent hierarchies, etc, are all issues in affective computing (and psychology for that matter).
- The proposed method is indeed scalable and can be replicated to extend the proposed dataset or correct the labels of existing benchmarks.
- With the rise of MLLMs, there is a growing interest in applying them to downstream tasks; however, the lack of annotations in natural language makes things more complicated, which is something this method addresses.

**Weaknesses:**

- One major weakness is the contradiction in motivation. Specifically, two of the key issues identified in the intro are a) MLLM's inaccurate responses when compared to human judgment and b) majority voting to address disagreement. The method then proceeds to create annotations by taking the majority voting of MLLMs. This is reiterated in the discussion (lines 448-9): "MLLMs may not yet be sufficiently competent for LLM-as-a-judge [...]". If this is the case, how are they sufficient as annotators? The method is contradictory to the problem it is trying to solve and thus is not convincing in terms of annotation accuracy. The one human expert curating statements post-hoc is probably not enough, as: a) there is already unconscious bias in the process as the expert is shown annotations rather than producing annotations, b) the task is uncertain even for experts, therefore, is one person really enough?, c) the method does not account for cultural differences and biases (both in the expert and the process as a whole). We also don't have an indication of how the humans (expert or otherwise) perform on the open-vocabulary compared to the LLMs. In general, I am unconvinced about the validity and motivation of the method.
- Looking at the human refinement process, the annotator agreement in Table 3 does not seem very strong, particularly for emotion interpretation and subjectivity, showing a (Cohen's?) Kappa of only 0.5. In fact, the only aspect where the human annotators agree is the scene context; for everything else, there is only weak agreement.
- For the fine-tuned results in Table 6, only 10k out of 462k images are selected. This seems a bit of an odd choice, as most large-scale affective datasets (including EmoSet) are much larger; therefore, it seems unlikely that the finetuned version will be able to compete, but at least performance gains from fine-tuning are shown.

**Questions:**

- How do the annotations generated by the proposed method in step [a] compare to the ones from EmoSet for the subset used in this work? Are the generated emotions in agreement with the 8 categorical used there?
- Using a small subset, is it possible to compare human open-vocabulary to MLLM open-vocabulary annotations? Are these in agreement or at least in the same region?
- The polarity underperformance may be due to the prompting used. Have you tried CoT reasoning? Polarity is a derivative, not a primary label, hence the suggestion.

---

> ### Author Response · Authors · 2025-11-20
> **Response to Reviewer ck1T (1/4)**
>
> Dear reviewer,
>
> Thank you for your detailed review and valuable feedback! We truly appreciate your time and the opportunity to address the concerns you raised. Below, we provide individual explanations for the weaknesses and questions raised, and we would be more than happy to further clarify any points that remain insufficiently addressed.
>
> >**Weakness 1 (1). One major weakness is the contradiction in motivation. Specifically, two of the key issues identified in the intro are a) MLLM's inaccurate responses when compared to human judgment and b) majority voting to address disagreement. The method then proceeds to create annotations by taking the majority voting of MLLMs. This is reiterated in the discussion (lines 448-9): "MLLMs may not yet be sufficiently competent for LLM-as-a-judge [...]". If this is the case, how are they sufficient as annotators? The method is contradictory to the problem it is trying to solve and thus is not convincing in terms of annotation accuracy.**
>
> Thank you for this thoughtful comment. In response, we would like to begin by clarifying the two issues mentioned in the introduction and explaining why they do not contradict the use of MLLMs in our annotation pipeline.
>
> First, regarding "*(a) MLLMs' inaccurate responses compared to human judgment*", **we would like to clarify that we did not make this statement in the form presented by the reviewer in the introduction**. Our exact motivation, stated in lines 38–42 of the manuscript, is that "*conventional emotion evaluation approaches are not compatible with MLLMs, leaving the emotional perception ability of MLLMs insufficiently agreed*". While we do acknowledge that MLLMs possess inherent inaccuracies (as later demonstrated in our evaluation results), we do not claim that their judgments are uniformly unreliable or unsuitable for all forms of emotional data construction.
>
> Second, this acknowledgment does not conflict with our use of MLLMs in the data construction pipeline. **Importantly, our method does not rely on MLLMs to directly produce final emotional statements. Instead, we intentionally restrict their role in generating preliminary results**, such as open-vocabulary emotion candidates and prototype statements, that support diversity and scalability while avoiding direct dependence on their subjective interpretations. **Building upon these preliminary outputs, we explicitly apply emotion-theoretic grounding and carefully designed construction rules to assign appropriate open-vocabulary emotions and produce image–statement pairs with minimal ambiguity. Most critically, we incorporate a necessary human refinement stage to correct errors and verify annotation quality, ensuring the reliability of the final benchmark.** Notably, similar uses of MLLMs as emotional data constructors have been validated in prior work, such as Emotion-Llama [1] (NeurIPS 2024) and AffectGPT [2] (ICML 2025), as also noted in lines 38-42 of the manuscript.
>
> Third, regarding "*(b) majority voting to address disagreement*", we fully agree that majority voting helps improve label reliability, as labels endorsed by multiple independent voters are more likely to reflect widely accepted interpretations. It is precisely for this reason that we adopt majority voting among MLLMs as **one component** of our strategy for mitigating the inherent inaccuracies of MLLMs.
>
> Taken together, we believe that MLLMs are suitable for generating preliminary results in data construction. While these outputs may contain inaccuracies, they can be sufficiently corrected through the subsequent strategies. Majority voting can enhance the reliability of these outcomes rather than replacing the need for additional safeguards. **Therefore, we respectfully disagree with the reviewer’s critiques that our use of MLLMs is contradictory to our motivation. We believe it denies the constructive role that MLLMs can play in preliminary annotation and overlooks the set of complementary strategies we introduce to mitigate their inaccuracies.**
>
> **References**
>
> [1]. Emotion-Llama: Multimodal Emotion Recognition and Reasoning with Instruction Tuning. NeurIPS 2024. \
> [2]. AffectGPT: A New Dataset, Model, and Benchmark for Emotion Understanding with Multimodal Large Language Models. ICML 2025.

---

> > ### Author Response · Authors · 2025-11-20
> > **Response to Reviewer ck1T (2/4)**
> >
> > >**Weakness 1 (2). The one human expert curating statements post-hoc is probably not enough, as: a) there is already unconscious bias in the process as the expert is shown annotations rather than producing annotations, b) the task is uncertain even for experts, therefore, is one person really enough?, c) the method does not account for cultural differences and biases (both in the expert and the process as a whole).**
> >
> > Thank you for raising this concern. In fact, **we incorporated five human annotators, instead of one, to refine the constructed statements through a majority-voting procedure**, as described in Lines 336–349 of the manuscript. Each image-statement pair is retained only if at least four annotators reach a consensus on its label. Through verification, 90.6% of automatically assigned labels are validated as accurate, 6.9% are identified as inaccurate, and 2.5% are discarded due to a lack of consensus. This process not only guarantees the high quality of final labels but also substantiates the inherent reliability of our automated pipeline.
> >
> > Regarding point (c), we agree that cultural differences and annotator-specific biases are valid and important concerns. Emotion perception is inherently subjective, and these differences can shape annotation outcomes. However, such factors remain a significant challenge for emotion-related benchmarks in general and, to the best of our knowledge, have not been satisfactorily addressed. Therefore, **one of the motivations behind our ESJ task reformulation is precisely to mitigate subjectivity at the task design level.** Under such task formulation, different viewers are more likely to provide consistent judgments, thereby enhancing the reliability of MLLM evaluation. However, we acknowledge that our original manuscript did not discuss this aspect with sufficient depth. In the revision, **we have substantially expanded the Ethics Statement section (Lines 488-528) to discuss such biases and facilitate responsible use of the benchmark.**
> >
> > >**Weakness 1 (3). We also don't have an indication of how the humans (expert or otherwise) perform on the open-vocabulary compared to the LLMs. In general, I am unconvinced about the validity and motivation of the method.**
> >
> > We kindly refer you to our response to **Question 2**, where we provide a detailed clarification of this concern. We hope that our explanations for both the identified weaknesses and the corresponding questions help clarify the validity and motivation of our method. We would be more than happy to further discuss any remaining issues or provide additional analyses if needed.
> >
> > >**Weakness 2. Looking at the human refinement process, the annotator agreement in Table 3 does not seem very strong, particularly for emotion interpretation and subjectivity, showing a (Cohen's?) Kappa of only 0.5. In fact, the only aspect where human annotators agree is the scene context; for everything else, there is only weak agreement.**
> >
> > Thank you for pointing this out. We would like to clarify that the reported coefficients are **Fleiss' Kappa**, which is known to be more conservative than Cohen's Kappa because it evaluates agreement among multiple annotators rather than between pairs. In [1], **Fleiss’ Kappa values in the range of 0.41–0.60 are commonly interpreted as "moderate agreement".** Given that the inherent higher level of subjectivity in the emotion interpretation and subjectivity perception dimensions, we consider 0.5 strong enough to reflect inter-annotater agreement and the effectiveness of mitigating subjectivity of ESJ. We apologize for this confusion and have clarified this issue in the updated manuscript.
> >
> > **References**
> >
> > [1]. The Measurement of Observer Agreement for Categorical Data. Biometric 1977.

---

> > > ### Author Response · Authors · 2025-11-20
> > > **Response to Reviewer ck1T (3/4)**
> > >
> > > >**Weakness 3. For the fine-tuned results in Table 6, only 10k out of 462k images are selected. This seems a bit of an odd choice, as most large-scale affective datasets (including EmoSet) are much larger; therefore, it seems unlikely that the finetuned version will be able to compete, but at least performance gains from fine-tuning are shown.**
> > >
> > > We appreciate you raising this point and fully understand the underlying concern. However, we would like to clarify that **the purpose of this experiment is not to develop an optimized method tailored for the MVEI benchmark or the ESJ task, but rather to provide further exploratory evidence that sheds light on MLLMs’ visual emotion intelligence**.
> > >
> > > In contrast, most large-scale affective datasets (such as EmoSet) are inherently designed for the fine-tuning–evaluation paradigm, where expert models are trained to solve a narrowly defined task (e.g., mapping images to one of eight predefined emotion categories). These datasets thereby require large-scale fine-tuning to internalize both the task definition, input-output mappings, and dataset-specific biases. Recognizing the incompatibility between such paradigms and MLLMs, **we proposed the ESJ task specifically as a more reliable zero-shot evaluation framework**, rather than replicating the fine-tuning–evaluation paradigm commonly used for smaller models.
> > >
> > > On the other hand, **the current fine-tuning scale is sufficient to reveal meaningful dimension-level characteristics of MLLMs**. As discussed in lines 461–467 of the manuscript, the improvement in the sentiment polarity dimension is substantial, whereas the gain in perception subjectivity is modest. This suggests that the former is primarily driven by confusion among positive, negative, and mixed categories, which can be alleviated with few-shot demonstrations or lightweight fine-tuning, while the latter reflects a more fundamental challenge likely requiring subjectivity-oriented pre-training objectives or dedicated datasets.
> > >
> > > >**Question 1. How do the annotations generated by the proposed method in step [a] compare to the ones from EmoSet for the subset used in this work? Are the generated emotions in agreement with the 8 categories used there?**
> > >
> > > Thank you for the question. We agree that a cross-validation based on EmoSet labels can provide an intuitive evaluation for our automatically assigned open-vocabulary labels. Since EmoSet is built upon the Mikels model, which contains eight emotion categories (amusement, awe, contentment, excitement, anger, disgust, fear, and sadness), it is not naturally aligned with the Parrott hierarchical emotion model adopted in our manuscript.
> > >
> > > However, we note that except for awe, the remaining seven Mikels categories are distributed across different levels of Parrott’s hierarchy. To enable comparison, we map them to the primary-level emotions in Parrott’s model: amusement → joy, contentment → joy, excitement → joy, anger → anger, disgust → anger, fear → fear, sadness → sadness. Finally, using the open-vocabulary attachment obtained from our constructed POM, we map awe → surprise. Based on this mapping, **our analysis shows that the automatically assigned open-vocabulary labels of 97.3% of the 17,716 images in INSETS-462k overlap with their original EmoSet labels at the primary level of the Parrot Model.** This high level of consistency provides strong evidence for the validity of our automated annotation pipeline.

---

> > > > ### Author Response · Authors · 2025-11-20
> > > > **Response to Reviewer ck1T (4/4)**
> > > >
> > > > >**Question 2. Using a small subset, is it possible to compare human open-vocabulary to MLLM open-vocabulary annotations? Are these in agreement or at least in the same region?**
> > > >
> > > > Thank you for this constructive suggestion. We conducted this evaluation on a relatively small scale. We sampled 50 images from MVEI and asked five graduate students to provide open-vocabulary emotion labels describing the emotions evoked when viewing each image. We observed that humans (possibly influenced by domain knowledge or descriptive habits) tend to produce far fewer distinct emotion terms compared with the diverse candidate labels generated by MLLMs. Therefore, for each image, we took the union of all human-provided emotions, resulting in an average of 3.7 human open-vocabulary emotion labels per image.
> > > >
> > > > To quantitatively compare human and MLLM annotations, we introduced two evaluation metrics. *Emotion Precision* is defined, for each image, as the proportion of MLLM labels that share the same primary-level Parrot emotion category with at least one human label. This measures the accuracy of MLLM labels relative to human perception. In contrast, *Emotion Recall* is defined as the proportion of human labels that share the same primary-level Parrot category with at least one MLLM label, capturing coverage of MLLM labels.
> > > >
> > > > The results show that the average *Emotion Precision* is 87.1%, and the *Emotion Recall* is 95.4% across 50 sampled images. The high recall indicates that the MLLM labels almost entirely cover the human labels. The relatively lower *Emotion Precision* reflects cases where the automatic pipeline produces MLLM labels not mentioned by the five annotators. We consider these not to be necessarily MLLMs' annotation errors. Instead, they might represent plausible emotional interpretations that were not articulated by the human annotators in this small-scale study. Although there is no direct baseline for comparison, **we reasonably consider both metrics to be high, offering additional evidence supporting the validity of our annotation pipeline**.
> > > >
> > > > >**Question 3. The polarity underperformance may be due to the prompting used. Have you tried CoT reasoning? Polarity is a derivative, not a primary label, hence the suggestion.**
> > > >
> > > > We sincerely appreciate this construction suggestion. By incorporating "*Think step-by-step*" into the instructions, we have conducted additional evaluation under the CoT reasoning setup. These new results are included in Table 6 of the updated manuscript and summarized in the table below.
> > > >
> > > > | Qwen2.5-VL (8.3B) | SP | EI | SC | PS | Total |
> > > > |---|:---:|:---:|:---:|:---:|---|
> > > > | Direct Inference | 63.2 | 81.5 | 83.9 | 66.3 | 75.9 |
> > > > | CoT Reasoning | 67.4 | 81.5 | 84.6 | 67.0 | 76.6 |
> > > >
> > > > As shown, the CoT setting indeed leads to a marked improvement in the sentiment polarity dimension, supporting the hypothesis that polarity can benefit from explicit reasoning. We also observe consistent improvements across other dimensions, indicating that CoT prompting helps the model generate more accurate affective judgments.

---

> > ### Comment · Reviewer_ck1T · 2025-11-20
> >
> > Thank you very much for the detailed responses! I will try to respond to all in one comment and keep the discussion going.
> > - one human expert: can you update the manuscript in that area, to clarify? it reads as if there is a single human expert and then a panel of 5 non-experts. With your explanation in the rebuttal the method is more understandable and of course addresses my initial points of subjectivity etc.
> > - Kappa: thanks for clarifying and updating the manuscript, indeed for Fleiss' kappa values .5 is in the moderate agreement area.
> > - Tab 6: I understand the aim of the work being exploratory, with that in mind for evaluations to be meaningful we need to have a common baseline. With that in mind, I am still unsure regarding using only a subset as we cannot compare with previous works (and any future work will also be unable to compare results fairly).
> > - Annotations compared to EmoSet: thanks for this comparison! I think it would be useful to include in the manuscript (if not already there).
> > - Q2-3: the results indeed looks promising and the human annotation evaluation is indeed convincing.
> > - weakness 1: indeed this is not how it is written in the paper, but given the review character limitations the points were abbreviated. There is an argument that can be made there regarding the reliability of LLMs even under majority voting: given that all have been trained on similar data chances are they have the same biases. With that in mind, particularly with the human agreement results provided by the authors, even if the motivation may seem a little contradictory (to me at least)  there seems to be merit in the annotation method.
> >
> > I remain unconvinced about the first weakness raised, i.e. the motivation, and the experiments in Tab 6. However, the clarifications from the authors and additional validations regarding alignment of annotations with human experts are indeed good, therefore I'm willing to raise the initial score.

---

> > > ### Author Response · Authors · 2025-11-23
> > > **Further Response to Reviewer ck1T (1/2)**
> > >
> > > Dear Reviewer,
> > >
> > > We sincerely thank you for the effort in reviewing our responses and for providing further constructive feedback. Your continued engagement has been invaluable in strengthening the manuscript. We address the remaining concerns below, and we would be glad to further clarify any points that may still require discussion. A new version of the manuscript has been updated.
> > >
> > > >**One human expert: Can you update the manuscript in that area, to clarify? it reads as if there is a single human expert and then a panel of 5 non-experts. With your explanation in the rebuttal the method is more understandable and of course addresses my initial points of subjectivity etc.**
> > >
> > > Thanks for this feedback. We acknowledge that our actual procedure is indeed as the manuscript reads: one expert was responsible for refining the attachment from open-vocabulary labels to the Parrott model, and a panel of five graduate students (not specialized in psychology) performed the voting-based verification of statement labels. In our earlier understanding of your comments (Weakness 1 (2)), we interpreted your concern as primarily the number of annotators (as stated: "is one person really enough?"). Accordingly, our rebuttal focused on clarifying that "we incorporated five human annotators, instead of one".
> > >
> > > Regarding annotator qualifications, we fully agree that graduate students not specialized in psychology do not possess the same level of training as a psychology postgraduate. However, **we would also like to clarify that we believe they are sufficiently capable of the task they were assigned**. In contrast to the process of refining open-vocabulary attachments, which requires a nuanced understanding of subtle emotional distinctions, verifying the correctness of emotional statements is a comparatively simpler judgment task. This can be intuitively supported by the high human accuracy reported in Table 5 of the manuscript. **Similar annotator choices are also common in prior affective datasets.** For example, EMOTIC (TPAMI 2020) [1] and ArtEmis (CVPR 2021) [2] explicitly stated that they recruited annotators from AMT, whose background is often not more specialized than graduate students.
> > >
> > > **References**
> > >
> > > [1]. Context Based Emotion Recognition using EMOTIC Dataset. TPAMI 2020. \
> > > [2]. ArtEmis: Affective Language for Visual Art. CVPR 2021.
> > >
> > >
> > > >**Tab 6: I understand the aim of the work being exploratory, with that in mind for evaluations to be meaningful we need to have a common baseline. With that in mind, I am still unsure regarding using only a subset as we cannot compare with previous works (and any future work will also be unable to compare results fairly).**
> > >
> > > Thank you for reiterating this important point. We agree that large-scale fine-tuning can indeed serve as baselines for broader comparisons. In light of this, we have incorporated additional experiments, including expanded LoRA fine-tuning, full-parameter fine-tuning, GRPO without CoT, and GRPO with CoT. For these experiments, we used the full INSETS-462K (excluding samples in MVEI) for supervised fine-tuning and a 50k subset for reinforcement learning (due to the high time cost). These results are summarized in the table below and **have been added to Appendix H of the updated manuscript**.
> > >
> > > | Qwen2.5-VL (8.3B) | #Sample | SP | EI | SC | PS | Total |
> > > |---|:---:|:---:|:---:|:---:|:---:|:---:|
> > > | Direct Inference | - | 63.2 | 81.5 | 83.9 | 66.3 | 75.9 |
> > > | CoT Reasoning | - | 67.4 | 81.5 | 84.6 | 67.0 | 76.6 |
> > > | LoRA  | 10k | 78.6 | 84.7 | 86.3 | 70.3 | 80.7 |
> > > | LoRA | 459k | 82.2 | 86.0 | 86.9 | 71.9 | 82.2 |
> > > | Full Fine-Tuning | 10k | 84.3 | 84.8 | 87.0 | 71.1 | 81.9 |
> > > | Full Fine-Tuning | 459k | 85.6 | 86.5 | 87.6 | 73.3 | 83.4 |
> > > | GRPO (without CoT) | 10k | 83.2 | 82.5 | 86.5 | 71.1 | 80.7 |
> > > | GRPO (without CoT) | 50k | 84.0 | 82.7 | 86.3 | 71.4 | 80.9 |
> > > | GRPO (with CoT) | 10k | 86.2 | 82.9 | 86.6 | 72.3 | 81.6 |
> > > | GRPO (with CoT) | 50k | 86.8 | 83.0 | 87.2 | 72.7 | 82.0 |
> > >
> > >
> > > >**Annotations compared to EmoSet: thanks for this comparison! I think it would be useful to include in the manuscript (if not already there).**
> > >
> > > Thank you for this suggestion. **We have incorporated this analysis into Appendix G of the updated manuscript**. We agree that it provides an intuitive validation of our proposed pipeline and meaningfully strengthens the manuscript.

---

> > > > ### Author Response · Authors · 2025-11-23
> > > > **Further Response to Reviewer ck1T (2/2)**
> > > >
> > > > >**Weakness 1: Indeed this is not how it is written in the paper, but given the review character limitations the points were abbreviated. There is an argument that can be made there regarding the reliability of LLMs even under majority voting: given that all have been trained on similar data chances are they have the same biases. With that in mind, particularly with the human agreement results provided by the authors, even if the motivation may seem a little contradictory (to me at least) there seems to be merit in the annotation method.**
> > > >
> > > > Thank you for this insightful argument regarding the potential issue of shared biases among MLLMs due to overlapping training data, which could indeed lead to a consensus on incorrect emotions. **However, we would like to clarify that, while this concern is valid in principle, it does not appear to be significant in practice based on our empirical observations.** As shown in Table 1 of the manuscript, the average number of open-vocabulary emotions extracted by different MLLMs varies considerably, ranging from 7.2 to 12.6. This demonstrates meaningful diversity among MLLMs. This diversity is also reflected in the Positive Ratio during evaluation: in the upper half of Table 4, the MLLMs involved in data construction exhibit a wide range, from 18.4 to 84.4, further indicating heterogeneous characteristics across MLLMs.
> > > >
> > > > On the other hand, we fully agree that MLLM-generated annotations, even with partial human involvement, cannot be expected to fully match or exceed the reliability of carefully curated human annotations. **However, MLLMs offer substantial advantages in annotation cost and scalability.** Although this scalability is reflected only as data scale within the current manuscript, the same benefit could extend to other languages, other image domains, and even other modalities. More importantly, through human verification, we show that the subsequent steps in our pipeline effectively mitigate MLLM inaccuracies and biases.
> > > >
> > > > **Taken together, and considering our core motivations, which include reliability, open-vocabulary coverage, multifaceted evaluation, and scalability, we believe the proposed tasks and pipeline achieve a meaningful balance that satisfactorily approaches all these objectives.**

---

> ### Comment · Reviewer_ck1T · 2025-11-25
>
> Thanks again for the clarifications! The concerns regarding experiments are addressed with the additional results provided.
>
> Regarding the annotation process and motivation, indeed [1,2] use similar process; however, [1] specifically has known issues with the annotations [3,4] which is partially where the criticism stems.
>
> Regarding the motivation, I appreciate the additional explanation; the annotation cost and scalability do indeed make more sense as motivation. I still believe there is a risk in using MLLMs for the annotation that is not fully acknowledged, but the explanation is understandable.
>
> I have raised my score.
>
> [3] Costa, et al. 2023. "A Survey on Datasets for Emotion Recognition from Vision: Limitations and In-the-Wild Applicability" Applied Sciences
>
> [4] Martínez-Miwa, et al. 2023  "On reliability of annotations in contextual emotion imagery." Scientific Data (nature)

---

> > ### Author Response · Authors · 2025-11-26
> > **Response to Reviewer ck1T**
> >
> > Once again, we sincerely thank you for this valuable discussion. Your feedback has greatly strengthened the manuscript. We also appreciate your insightful suggestion regarding annotation qualification, which we will carefully consider in future works.

---

### Official Review · Reviewer_Kehu · 2025-10-26

**Soundness:** 3
**Presentation:** 2
**Contribution:** 2
**Rating:** 4
**Confidence:** 4

**Summary:**

This paper introduces a new evaluation paradigm for assessing multimodal large language models (MLLMs) on visual emotion understanding. Instead of traditional emotion classification, the authors propose an Emotion Statement Judgment (ESJ) task, where models evaluate whether a given emotion-related statement about an image is correct. The paper further presents an automated pipeline (INSETS) to construct large-scale emotion-centric statements and builds the MVEI benchmark covering four dimensions: sentiment polarity, emotion interpretation, scene context, and perception subjectivity. Experiments show that current MLLMs demonstrate certain capabilities but still lag behind humans, especially in subjective and contextual emotional reasoning.

**Strengths:**

Comprehensive emotional dimensions, covering aspects that existing benchmarks ignore, such as scene context and perception subjectivity.

Extensive experiments on a wide range of MLLMs with meaningful insights and human comparison results.

**Weaknesses:**

The dataset is partially constructed using existing MLLMs, which may introduce inherited biases. Although human refinement is applied, its proportion remains limited.

The distinction from EEmo-Bench: A Benchmark for Multi-modal Large Language Models on Image Evoked Emotion Assessment is not sufficiently articulated.

The dataset is built on a single source domain, and its generalizability to other visual contexts is unclear.

**Questions:**

Please refer to the weakness.

---

> ### Author Response · Authors · 2025-11-20
> **Response to Reviewer Kehu (1/2)**
>
> Dear reviewer,
>
> Thank you for your detailed review and valuable feedback! We truly appreciate your time and the opportunity to address the concerns you raised. Below, we provide individual explanations for the weaknesses and questions raised, and we would be more than happy to further clarify any points that remain insufficiently addressed.
>
> >**Weakness 1 (1). The dataset is partially constructed using existing MLLMs, which may introduce inherited biases.**
>
> We fully acknowledge this concern. However, we would like to clarify that our evaluation was intentionally structured to avoid this issue in the original manuscript.
>
> To ensure fairness, **we explicitly segregated the MLLMs used to construct INSETS from those that were not**. As stated in the caption of Table 4 of the manuscript: "*For fair comparison, we separate the MLLMs involved in constructing INSETS-462k (the upper part) and the others (the lower part).*" and in the caption of Table 5: "*To ensure fairness, the results of partial leading MLLMs are also reported, adhering to the same partition as Table 4.*"
>
> Correspondingly, **our analysis also explicitly avoided any direct performance comparison between the two groups**, and placed greater emphasis on the non-constructing MLLMs. We believe that this separation ensures that the benchmark results are not inflated and maintains the independence of the evaluation.
>
>
> >**Weakness 1 (2). Although human refinement is applied, its proportion remains limited.**
>
> We completely agree that the expansion of human-refined data comes with improved evaluation stability and reliability. However, **we respectfully point out that, even at its current scale, the benchmark already yields highly consistent and reliable evaluation outcomes**. Due to the formulation of Emotion Statement Judgement, the inherent ambiguity of each query is significantly reduced, leading to highly consistent model outputs across different trials.
>
> | Model | SP | EI | SC | PS |
> |---|:---:|:---:|:---:|:---:|
> | DeepSeek-VL | 68.7 (+-0.2) | 70.8 (+-0.3) | 81.1 (+-0.1) | 73.2 (+-0.3) |
> | MiniCPM | 70.4 (+-0.4) | 78.4 (+-0.2) | 81.9 (+-0.3) | 70.5 (+-0.3) |
> | Qwen2.5-VL | 63.2 (+-0.3) | 81.5 (+-0.5) | 83.9 (+-0.1) | 66.3 (+-0.2) |
> | GPT4o | 72.5 (+-0.2) | 84.3 (+-0.3) | 81.6 (+-0.2) | 69.2 (+-0.3) |
>
> The table above reports the results from splitting the three queries per sample into three separate evaluations. The results reveal that the variance across evaluations is consistently minimal across four dimensions (SP: Sentiment Polarity, EI: Emotion Interpretation, SC: Scene Context, PS: Perception Subjectivity), validating the stability and reliability at the current evaluation scale. Notably, a similar or even smaller scale is also prevalent in some recent multimodal benchmarks. For instance, HallusionBench [1] (CVPR 2024) contains 1,129 QA pairs, and MMAR [2] (NeurIPS 2025) includes 1,000 QA pairs.
>
> Moreover, **our proposed INSETS pipeline is designed to be scalable**. This enables labour-efficient expansion, whether by extending annotations to additional data within INSETS-462k or by applying the pipeline to new image sources to synthesize further samples.
>
> **References**
>
> [1] HallusionBench: An Advanced Diagnostic Suite for Entangled Language Hallucination and Visual Illusion in Large-Language Models. CVPR 2024. \
> [2] MMAR: A Challenging Benchmark for Deep Reasoning in Speech, Audio, Music, and Their Mix. NeurIPS 2025.

---

> > ### Author Response · Authors · 2025-11-20
> > **Response to Reviewer Kehu (2/2)**
> >
> > >**Weakness 2. The distinction from EEmo-Bench: A Benchmark for Multi-modal Large Language Models on Image Evoked Emotion Assessment is not sufficiently articulated.**
> >
> > Thank you for pointing us to this highly relevant and well-designed work. Based on a careful review, we would like to clarify the distinctions as follows.
> >
> > In EEmo-Bench, the authors aim to evaluate the evoked emotion understanding abilities of MLLMs through four specialized tasks: Perception, Ranking, Description, and Assessment. A notable innovation is the introduction of multi-image inputs, which increases task complexity and significantly deepens the evaluation scope.. Furthermore, by incorporating manually annotated emotion rankings and VAD (Valence, Arousal, Dominance) value, the benchmark broadens the coverage of existing visual emotion evaluation for MLLMs. These insightful designs are highly inspiring.
> >
> > Despite its significant contributions, we observe that **EEmo-Bench does not fully avoid the ambiguity inherent in open-ended responses to the extent achieved by our proposed Emotional Statement Judgement (ESJ) task**. Additionally, it also **falls short specifically with respect to the three advantages emphasized in our work, namely, open-vocabulary coverage, multifaceted evaluation, and scalability**.
> >
> > First, to our understanding, most of the What-How questions in EEmo-Bench's Perception subset are constructed based on a single fixed emotion category or VAD value. Although VAD allows for continuous emotion representation, it inherently reflects only a single perspective of an image, whereas VAD ratings can vary significantly across individuals. **This can lead to multiple-choice questions where plausible answers are deemed incorrect, an issue largely mitigated by our ESJ formulation**. A similar concern also appears in the Ranking and Assessment subsets. Despite their unique strengths, their broadened task coverage inevitably increases the potential for ambiguity.
> >
> > Second, the emotion categories in EEmo-Bench are restricted to a relatively narrow set of seven classes. **In contrast, the proposed MVEI benchmark employs an open-vocabulary labeling approach**, which significantly enhances the granularity of evaluation and better challenges MLLMs' understanding of diverse emotional expressions.
> >
> > Third, EEmo-Bench primarily focuses on emotion recognition, with a modest extension toward emotion interpretation, and does not include the scene context or perception subjectivity dimensions. Consequently, while our benchmark may be less deep in recognition alone, **it integrates both intrinsic image attributes and contextual factors essential to human emotional perception, establishing a more holistic evaluation framework.**
> >
> > Fourth, EEmo-Bench heavily relies on manual annotation of emotion categories, emotion rankings, and VAD values, whereas **our INSETS pipeline offers a highly automated approach to data construction**. This makes our framework more scalable and labour-efficient.
> >
> > We would like to once again express our appreciation to you for highlighting this important work. We commit to thoroughly discussing EEmo-Bench in future versions of the manuscript based on the above analysis (we have temporarily not  incorporated this discussion into the current version due to page constraints during rebuttal and the broader structural adjustments that a revision of the related work section would require).
> >
> > >**Weakness 3. The dataset is built on a single source domain, and its generalizability to other visual contexts is unclear.**
> >
> > Thank you for raising this important concern. We would like to clarify that our pipeline is highly automated: it operates directly on **raw, unlabeled images** to generate open-vocabulary emotion labels and constructs corresponding emotional statements. This process can, in principle, be seamlessly transferred to any image collection.
> >
> > At the same time, we acknowledge the possibility of domain sensitivity arising from the manual refinement of open-vocabulary emotion attachment. However, **we believe the associated human cost remains largely manageable**. As shown in Table 2 of the manuscript, when the total number of statements increases from 3k to 462k (a 154× expansion), the number of distinct open-vocabulary emotion labels grows by only 77% (from 424 to 751). This growth reflects the natural boundedness of human emotional expression, indicating that the refinement workload expands at a decelerating rate. Taken together, these observations suggest that the proposed pipeline can generalize to different image domains, with only manageable labor overhead.

---

### Official Review · Reviewer_joCJ · 2025-10-29

**Soundness:** 2
**Presentation:** 2
**Contribution:** 2
**Rating:** 4
**Confidence:** 3

**Summary:**

This paper addresses the incompatibility of conventional emotion evaluation methods with Multimodal Large Language Models (MLLMs) by proposing the Emotion Statement Judgment (ESJ) task and the INSETS annotation pipeline. The work constructs MVEI, a benchmark with 3,086 samples covering four dimensions: sentiment polarity, emotion interpretation, scene context, and perception subjectivity. The authors evaluate 18 MLLMs and demonstrate that even top-performing models like GPT4o fall substantially short of human performance.

**Strengths:**

1. Benchmark Innovation and Scope: The ESJ task and MVEI benchmark fill a substantial gap in visual emotion evaluation for MLLMs, moving beyond rigid classification to nuanced, open-vocabulary, and multifaceted judgment tasks. This approach allows broader, context-aware, and fine-grained assessment.
2. Automated scalable pipeline: The INSETS pipeline effectively reduces human labor by automating emotion tagging and statement generation while maintaining a reported 90.6 % accuracy after human refinement.
3. Comprehensive empirical analysis: The benchmark evaluation spans >15 MLLMs, both open and proprietary, and compares against human baselines .
4. Grounding in Theory: The mapping to Parrott’s hierarchical emotion model and the use of extensive prompts/protocols demonstrate that the benchmark is theoretically informed, not ad-hoc.

**Weaknesses:**

1. Potential circularity in evaluation: Some MLLMs used to construct INSETS also participate in evaluation, which could inflate results and reduce benchmark independence.
2. Vague or Underspecified Mathematical/Labeling Procedures: The emotion label assignment, majority voting, and mapping to the hierarchical model are described mainly in prose and schematic diagrams, with some reliance on prompt outputs and “manual expert check.” The specific algorithms are insufficiently formalized.
3. Human annotation reliability and bias: Although inter-annotator agreement statistics are reported, there is limited qualitative analysis of ambiguous cases or bias due to annotator demographics.
4. Insufficient discussion of dataset bias and ethical risk: Bias acknowledgement is limited to a short ethics statement without empirical analysis of representation balance (e.g., emotional polarity, demographic context).
5. Limited analysis of failure modes: Sec. 5.3 identifies that "perception subjectivity shows only modest improvement"  but provides no error analysis. What types of subjectivity statements are most challenging? Are errors systematic? Figure 12 shows examples but no quantitative breakdown.
6. Absence of ablation or component analysis: There is no direct evidence quantifying how each component contributes to final benchmark quality.

**Questions:**

Please check the weakness part.

---

> ### Author Response · Authors · 2025-11-20
> **Response to Reviewer joCJ (1/4)**
>
> Dear reviewer,
>
> Thank you for your detailed review and valuable feedback! We truly appreciate your time and the opportunity to address the concerns you raised. Below, we provide individual explanations for the weaknesses and questions raised, and we would be more than happy to further clarify any points that remain insufficiently addressed.
>
> >**Weakness 1.  Potential circularity in evaluation: Some MLLMs used to construct INSETS also participate in evaluation, which could inflate results and reduce benchmark independence.**
>
> We fully acknowledge this concern. However, we would like to clarify that our evaluation was intentionally structured to avoid this issue in the original manuscript.
>
> To ensure fairness, **we explicitly segregated the MLLMs used to construct INSETS from those that were not**. As stated in the caption of Table 4 of the manuscript: "*For fair comparison, we separate the MLLMs involved in constructing INSETS-462k (the upper part) and the others (the lower part).*" and in the caption of Table 5: "*To ensure fairness, the results of partial leading MLLMs are also reported, adhering to the same partition as Table 4.*"
>
> Correspondingly, **our analysis also explicitly avoided any direct performance comparison between the two groups**, and placed greater emphasis on the non-constructing MLLMs. We believe that this separation ensures that the benchmark results are not inflated and maintains the independence of the evaluation.

---

> > ### Author Response · Authors · 2025-11-20
> > **Response to Reviewer joCJ (2/4)**
> >
> > >**Weakness 2.  Vague or Underspecified Mathematical/Labeling Procedures: The emotion label assignment, majority voting, and mapping to the hierarchical model are described mainly in prose and schematic diagrams, with some reliance on prompt outputs and “manual expert check.” The specific algorithms are insufficiently formalized.**
> >
> > We appreciate your insightful comment. For clarity, we note that majority voting and hierarchical mapping are sub-steps within the broader emotion label assignment. To address the concern holistically, we have structured our clarification around the three key sub-steps of emotion label assignment, providing additional details and formalisms where appropriate.
> >
> > **1. Extraction and filtering of open-vocabulary emotions.** \
> > These steps first extract potential open-vocabulary emotions of images using multiple MLLMs, and then filter out words unsuitable as emotion descriptors. They are essentially prompt-defined operations (with the full prompts specified in the Appendix) applied deterministically across MLLMs. Since these processes operate directly on natural language outputs, we believe a formal algorithmic description would not substantially enhance clarity beyond the textual explanation and the accompanying visual workflow in Fig. 2(a)(b).
> >
> > **2. Mapping open-vocabulary emotions onto the hierarchical model.** \
> > This step aims to attach the open-vocabulary emotions to Parrott’s hierarchical emotion model, which operates in two parts. First, GPT-4 is prompted to categorize each open-vocabulary term into the closest tertiary emotion in Parrott’s hierarchy. Similar to the previous, this process is also a prompt-defined operation. Second, a hired psychology expert conducts a manual refinement for the GPT-categorized map. We acknowledge that this part was not sufficiently elaborated in the original manuscript. In practice, the expert performed a straightforward, term-by-term review and adjusted assignments only when a more semantically appropriate match existed. This one-time process took approximately 15 hours and is efficient relative to the scale of the dataset.
> >
> > **3. Majority-voting mechanism.** \
> > Finally, this step selects consensus open-vocabulary emotion labels for images, thereby enhancing reliability. In contrast to the previous, this part indeed lends itself well to formalization. To enhance clarity and transparency, we provide the following formal definition.
> >
> > Let an image be processed by $n$ MLLMs. The set of open-vocabulary emotion labels generated by the $i$-th model is denoted by $L_i = \lbrace e_{i,1}, e_{i,2}, \dots, e_{i,m}\rbrace$, where $i\in \lbrace 1,2,\dots,n \rbrace $.
> > Let $\mathcal{P}$ be the mapping from an open-vocabulary emotion to the secondary emotion in POM. The ensemble-based majority-voting procedure is defined as follows.
> >
> > 1). Quota allocation.\
> >     For a secondary emotion $\bar{e}$, define its quota $Q(\bar{e})$ as:
> > \begin{equation}
> >     Q(\bar{e}) = \max\left(\left\lfloor \sum_{i=1}^{n} \mathbb{I}[\bar{e} \in \{\mathcal{P}(e) \mid e \in L_i\}] - \frac{n}{2} + 1 \right\rfloor, 0 \right)
> > \end{equation}
> >     where $\mathbb{I}[\cdot]$ is the indicator function. This quantity is strictly positive only when $\bar{e}$ is supported by a majority of the MLLMs.
> >
> > 2). Candidate Pool Formation.\
> >     The candidate pool of open-vocabulary labels for the secondary emotion $\bar{e}$ is defined as:
> > \begin{equation}
> > S(\bar{e}) = \bigl\lbrace e_{i,j} \bigm| i \in \lbrace 1,2,\dots,n \rbrace; j\in \lbrace 1,2,\dots,m \rbrace; \mathcal{P}(e_{i,j}) = \bar{e} \bigr\rbrace.
> > \end{equation}
> >
> > 3). Consensus Selection.\
> >     Let $\mathrm{freq}(e)$ denote the frequency of an open-vocabulary label $e$ in $S(\bar{e})$: $\mathrm{freq}(e)=|\lbrace e_{i,j}\in S(\bar{e})|e_{i,j}=e\rbrace|$.
> >     The consensus labels for $\bar{e}$ are defined as the top-$Q(\bar{e})$ unique labels in $S(\bar{e})$ ranked by $\mathrm{freq}(\cdot)$, where ties are resolved uniformly at random.
> >     The final consensus label set for the image is obtained by taking the union over all secondary emotions:
> > \begin{equation}
> >     L_{\mathrm{cons}} = \bigcup_{\bar{e}} \mathrm{Top}_{Q(\bar{e})}\left( S(\bar{e}) \right).
> > \end{equation}
> > This definition is updated in Appendix E of the manuscript. We sincerely appreciate your rigorous feedback, which has helped us significantly improve the methodological clarity and reproducibility of our work.

---

> > > ### Author Response · Authors · 2025-11-20
> > > **Response to Reviewer joCJ (3/4)**
> > >
> > > >**Weakness 3.  Human annotation reliability and bias: Although inter-annotator agreement statistics are reported, there is limited qualitative analysis of ambiguous cases or bias due to annotator demographics.**
> > >
> > > We agree that a deeper qualitative analysis of ambiguous cases can provide valuable insights beyond quantitative agreement statistics. In response to this comment, **we have now added a qualitative analysis of ambiguous cases identified during the human refinement process in Appendix F of the updated manuscript**.
> > >
> > > This analysis examines three representative samples that were excluded from the MVEI benchmark due to ambiguity. Based on this examination, we conclude that these ambiguities can be attributed to two principal factors: **the inherent limitations of MLLMs in visual perception**, which manifest as inaccurate descriptions or superficial analyses, and **the intrinsic challenges of emotion-related data construction**, where contextual subjectivity can render multiple interpretations valid. These observations collectively highlight the critical necessity of human refinement in data construction pipelines involving MLLMs.
> > >
> > > >**Weakness 4. Insufficient discussion of dataset bias and ethical risk: Bias acknowledgement is limited to a short ethics statement without empirical analysis of representation balance (e.g., emotional polarity, demographic context).**
> > >
> > > Thank you for raising this critical issue. In response, **we have significantly expanded our discussion of potential ethical risks**. To promote its responsible use, the updated manuscript now includes a detailed analysis spanning lines 488–528, covering:
> > > - The distributional imbalances in the emotional label of images.
> > > - The latent demographic biases that may persist in the perception subjectivity statements.
> > > - The potential cultural-perspective and aesthetic-perception biases of MLLMs.
> > > - The potential annotation-level biases due to cultural differences or personal experiences.
> > >
> > > Your suggestion has been invaluable in helping us strengthen the manuscript in this important dimension.
> > >
> > > >**Weakness 5. Limited analysis of failure modes: Sec. 5.3 identifies that "perception subjectivity shows only modest improvement" but provides no error analysis. What types of subjectivity statements are most challenging? Are errors systematic? Figure 12 shows examples but no quantitative breakdown.**
> > >
> > > Thank you for this insightful question. We agree that a deeper breakdown of challenging subjectivity types and potential systematic errors would indeed be valuable, yet it is currently beyond the reach of our framework. The immediate reason is that the characters involved in the subjectivity statements are produced without explicit structural control. As a result, **their demographic attributes cannot be meaningfully clustered, making it difficult to identify systematic error patterns quantitatively**.
> > >
> > > More fundamentally, the primary focus of this manuscript is to build a scalable evaluation framework for MLLMs that spans multiple affective dimensions, including sentiment polarity, emotion interpretation, scene context, and perception subjectivity. **Under this research scope, we design the data construction pipeline to generalize to the four broad dimensions, which inevitably sacrifices its compatibility with finer-grained evaluation**. Despite such trade-offs, current evaluations have led us to some critical and insightful conclusions about current MLLMs, as also kindly acknowledged by you in our strengths.
> > >
> > > We nonetheless recognize the importance of such fine-grained analyses and consider them a worthwhile direction for future exploration. Moreover, to ensure responsible and transparent use of the benchmark, we have explicitly incorporated this limitation, along with the potential for future improvement,  into the Ethical Statement of the updated manuscript.

---

> > > > ### Author Response · Authors · 2025-11-20
> > > > **Response to Reviewer joCJ (4/4)**
> > > >
> > > > >**Weakness 6. Absence of ablation or component analysis: There is no direct evidence quantifying how each component contributes to final benchmark quality.**
> > > >
> > > > We have not conducted ablation analysis for pipeline components primarily for two reasons.
> > > >
> > > > **First, the data construction pipeline is tightly coupled, and each component is essential to preserving the benchmark’s intended properties.** For instance, constraining open-vocabulary emotion labels to a fixed taxonomy would inherently remove the advantages of flexible and nuanced affect representation, and reducing the dimensions involved in statement generation would directly weaken the multifaceted nature of the benchmark. These effects are largely surface-level and structurally evident, making it less informative to perform ablations.
> > > >
> > > > **Second, quantitatively evaluating dataset quality itself is labour-intensive, yet the results are largely predictable.** In terms of construction accuracy, our human refinement process indicates that 90.6% of the constructed data is deemed correct, demonstrating the overall reliability of the pipeline. Although this human-based evaluation could, in principle, be used to ablate specific components (such as the number of MLLMs involved or the filtering of unqualified emotion words), the cost–benefit trade-off of such experiments is extremely high. We therefore opted for settings that are intuitive and empirically reasonable.
> > > >
> > > > If you have specific components or ablation directions you believe would be particularly insightful, we would be very happy to discuss and incorporate them in future iterations.

---

### Official Review · Reviewer_nM4y · 2025-11-01

**Soundness:** 3
**Presentation:** 3
**Contribution:** 3
**Rating:** 6
**Confidence:** 4

**Summary:**

This paper addresses the evaluation of visual emotional perception in Multimodal Large Language Models (MLLMs). The authors argue that prior evaluation methods—fixed label classification and coarse taxonomies—fail to capture the subjectivity, contextual dependence, and open-vocabulary nature of emotion perception. To remedy this, they propose the Emotion Statement Judgment (ESJ) task: MLLMs judge whether emotion-focused statements about an image are correct, enabling open-vocabulary, multifaceted assessment. They also introduce INSETS, an automated pipeline that generates emotion-centric labels and statement candidates with limited human intervention. Using INSETS they build a large automatically annotated corpus (INSETS-462k) and then refine a human-validated benchmark (MVEI) of 3,086 image–statement pairs covering four complementary dimensions: sentiment polarity, emotion interpretation, scene context, and perception subjectivity. Systematic evaluation of many MLLMs shows that models have notable strengths (especially in emotion interpretation and scene context) but still trail humans—particularly on sentiment polarity and perception subjectivity. The paper also explores lightweight adaptation methods (in-context learning, LoRA, full fine-tuning, GRPO) and finds that adaptation improves performance—most dramatically for sentiment polarity—while perception subjectivity remains challenging. The authors release code and data and discuss ethical considerations around dataset biases and model outputs.Three main contributions:1. Task formulation: Introduces the Emotion Statement Judgment (ESJ) task, a flexible, open-vocabulary framework that reframes visual emotion evaluation as a statement verification problem, reducing issues from rigid ground-truth answers and enabling richer, multifaceted assessment.2. Scalable annotation pipeline and corpus: Proposes INSETS, an automated pipeline for generating emotion labels and statements with minimal human effort, and uses it to construct INSETS-462k (462k statements across ~17.7k images), significantly improving scalability over prior labor-intensive datasets.3. Benchmark and empirical analysis: Curates MVEI (3,086 human-refined image–statement pairs) covering four evaluation dimensions (sentiment polarity, emotion interpretation, scene context, perception subjectivity), and provides systematic benchmarking of many contemporary MLLMs plus adaptation studies that reveal strengths, weaknesses, and directions for improvement.Three main shortcomings / limitations:1. Model scale coverage: The evaluation is limited mainly to MLLMs with fewer than ~10B parameters because of computational constraints, excluding larger open-source or closed models that might perform differently; this limits conclusions about upper-bound capabilities.2. Monolingual focus and potential biases: The current implementation and benchmark are monolingual (presumably English) and the automatically generated INSETS-462k may inherit biases from pretraining data and MLLMs; despite human refinement some problematic or culturally specific labels could persist.3. Fundamental difficulty with subjectivity: While adaptation improves sentiment polarity substantially, perception subjectivity remains poorly handled by MLLMs and shows only modest gains from fine-tuning; this suggests deeper architectural or objective-level limitations that the current pipeline and data do not fully resolve.

**Strengths:**

- Task formulation: Introduces the Emotion Statement Judgment (ESJ) task, a flexible, open-vocabulary framework that reframes visual emotion evaluation as a statement verification problem, reducing issues from rigid ground-truth answers and enabling richer, multifaceted assessment.
- Scalable annotation pipeline and corpus: Proposes INSETS, an automated pipeline for generating emotion labels and statements with minimal human effort, and uses it to construct INSETS-462k (462k statements across ~17.7k images), significantly improving scalability over prior labor-intensive datasets.
- Benchmark and empirical analysis: Curates MVEI (3,086 human-refined image–statement pairs) covering four evaluation dimensions (sentiment polarity, emotion interpretation, scene context, perception subjectivity), and provides systematic benchmarking of many contemporary MLLMs plus adaptation studies that reveal strengths, weaknesses, and directions for improvement.

**Weaknesses:**

- Model scale coverage: The evaluation is limited mainly to MLLMs with fewer than ~10B parameters because of computational constraints, excluding larger open-source or closed models that might perform differently; this limits conclusions about upper-bound capabilities.
- Monolingual focus and potential biases: The current implementation and benchmark are monolingual (presumably English) and the automatically generated INSETS-462k may inherit biases from pretraining data and MLLMs; despite human refinement some problematic or culturally specific labels could persist.
- Fundamental difficulty with subjectivity: While adaptation improves sentiment polarity substantially, perception subjectivity remains poorly handled by MLLMs and shows only modest gains from fine-tuning; this suggests deeper architectural or objective-level limitations that the current pipeline and data do not fully resolve.

**Questions:**

SEE WEAKNESS

---

> ### Author Response · Authors · 2025-11-20
> **Response to Reviewer nM4y**
>
> Dear reviewer,
>
> Thank you for your detailed review and valuable feedback! We truly appreciate your time and the opportunity to address the concerns you raised. Below, we provide individual explanations for the weaknesses and questions raised, and we would be more than happy to further clarify any points that remain insufficiently addressed.
>
> >**Weakness 1.  Model scale coverage: The evaluation is limited mainly to MLLMs with fewer than ~10B parameters because of computational constraints, excluding larger open-source or closed models that might perform differently; this limits conclusions about upper-bound capabilities.**
>
> We agree that comprehensive coverage is crucial for forming a complete picture of the visual emotion perception capabilities of MLLMs. While computational constraints prevented us from evaluating very large open-source models, **we deliberately incorporated assessments of advanced closed-source models in our original submission to help mitigate this limitation**. Specifically, as presented in Tables 4 and 5, we evaluated GPT4o and GPT4o-mini, two state-of-the-art, representative proprietary MLLMs available at the time of writing.
>
> Results in Table 4 indicate that although GPT-4o leads in overall performance, its advantage over other MLLMs remains relatively marginal. Notably, it underperforms in specific capabilities such as scene context and perception subjectivity. GPT-4o-mini achieves lower performance, roughly on par with leading open-source models. These observations are further reinforced in the human-comparison subset (Table 5), where all models exhibit a substantial performance gap compared to human participants.
>
> **Together, these findings suggest that the "upper-bound capabilities" of existing MLLMs in visual emotion perception are still far from saturated, highlighting considerable room for future improvement.**
>
> >**Weakness 2.  Monolingual focus and potential biases: The current implementation and benchmark are monolingual (presumably English), and the automatically generated INSETS-462k may inherit biases from pretraining data and MLLMs; despite human refinement, some problematic or culturally specific labels could persist.**
>
> Thank you for this constructive feedback. We acknowledge that the current implementation and benchmark are indeed English-only. However, **we would like to emphasize that the proposed pipeline itself is language-agnostic and could be adapted to other linguistic contexts with relatively minimal adjustments**, a point that we have also noted in the Limitations section of the paper.
>
> Furthermore, **we have significantly expanded our discussion of potential biases in the benchmark**. To promote its responsible use, the updated manuscript now includes a detailed analysis spanning lines 488–528, covering:
> - The distributional imbalances in the emotional label of images.
> - The latent demographic biases that may persist in the perception subjectivity statements.
> - **The potential cultural-perspective and aesthetic-perception biases of MLLMs**.
> - **The potential annotation-level biases due to cultural differences or personal experiences**.
>
> Your suggestion has been invaluable in helping us strengthen the manuscript in this important dimension.
>
> >**Weakness 3. Fundamental difficulty with subjectivity: While adaptation improves sentiment polarity substantially, perception subjectivity remains poorly handled by MLLMs and shows only modest gains from fine-tuning; this suggests deeper architectural or objective-level limitations that the current pipeline and data do not fully resolve.**
>
> We appreciate this thoughtful observation. The modest gains in the subjectivity perception dimension indeed point to a fundamental limitation of current MLLMs, which is precisely the kind of challenge our benchmark is designed to uncover.
>
> In fact, this empirical result is a key finding of our work. It serves to validate and quantify the deeper architectural limitations you mentioned. Therefore, **we view this not as a weakness of the pipeline, but as a contribution of our benchmark**: to definitively surface these potential challenges and guide future research toward solving them.

---

### Author Response · Authors · 2025-11-27
**Inquiry on Discussion Period Feedback**

Dear reviewers,

We are sincerely grateful for the thoughtful time and effort you have dedicated to evaluating our submission. With the discussion phase ending in a week, we want to kindly reach out to see whether our previous replies have adequately clarified the points you raised. Should there be any remaining issues that would benefit from further explanation, we would greatly appreciate receiving your feedback at your earliest convenience, so that we can respond promptly during the remaining discussion period.

---

### Author Response · Authors · 2025-12-01
**Summary of the Rebuttal Period**

We sincerely thank the reviewing committee for the time and effort invested in evaluating our work. Due to the unexpected incident of OpenReview, we would like to summarize the key points discussed before the rebuttal period was interrupted.

In this manuscript, we identify fundamental limitations in how existing approaches evaluate MLLMs’ visual emotion perception and introduce a new evaluation formulation, Emotion Statement Judgement, to mitigate these issues. Accordingly, we also design an automated annotation pipeline. Together, they support a customized, open-vocabulary, multifaceted, and scalable evaluation of MLLMs. Combined with labor-efficient human refinement, this pipeline produces the MVEI benchmark, which provides reliable feedback signals and insights into the visual emotional intelligence of MLLMs.

In the initial reviews, **our task formulation was explicitly considered valid by 3 out of 4 reviewers; our data construction pipeline was recognized as valuable by 3 out of 4 reviewers; and our evaluation results were regarded as comprehensive and insightful by all 4 reviewers**. Several reviewers also highlighted the generalizability of our annotation framework, our integration of emotion theory, and the substantial gap our work fills in evaluating MLLMs’ visual emotion intelligence.

Apart from the issues covered in the manuscript, the reviewers' primary concerns centered on whether the use of MLLMs in the annotation pipeline could introduce systematic biases, the need for additional quantitative and qualitative analysis of human refinement, and further discussion of annotator subjectivity and benchmark biases. **We have provided point-by-point responses to these highly constructive suggestions and incorporated the corresponding results into the updated manuscript**. These additions meaningfully improved the clarity of our presentation and strengthened the responsible-use guidance for the benchmark.

Until the rebuttal period was interrupted, one reviewer (ck1T) had responded to our clarifications. Their previous comments mainly concerned our motivation, the quantitative validation of the human-refinement process, and the performance of broader MLLM adaptation techniques. **Through an in-depth exchange, the reviewer indicated that their concerns had been resolved and that they intended to raise their score.** Although the remaining reviewers were unable to respond due to the outage, we remain cautiously confident that our responses would have addressed their concerns as well.

Despite the disruption, this rebuttal process has been highly valuable to us. We would like to once again extend our sincere appreciation to all reviewers for their thoughtful and constructive engagement.

---

### Meta-Review · Area_Chair_DWcu · 2026-01-08

**Summary:**

This is a borderline case.

The paper receives 4 high-quality reviews, with 3 negative initial ratings (4, 4, 4) and 1 positive initial rating (6).

Only 1 reviewer (rating: 4) has participated in the discussions, and after rebuttal the reviewer still has doubts using MLLMs for the annotation, but
the reviewer does accept annotation cost and scalability as motivation, and is willing to raise the rating.

Another 2 negative reviewers (rating: 4, 4) have not participated in the discussions. One (Reviewer Kehu) has concerns similar to the above reviewer. These 2 reviewers also have some other specific concerns, and the authors have given detailed responses and additional results for addressing these concerns.

The positive reviewer has also not participated in the discussions and the authors have given detailed responses and additional results.

**Reviewer Concerns:**

See the above summary.

**Reviewer Scores:**

See the above summary.

---

### Decision · Program_Chairs · 2026-01-26

Accept (Poster)